# Identifying and Evaluating Inactive Heads in Pretrained LLMs

**Pedro Sandoval-Segura**[1]    **Xijun Wang**[1]    **Ashwinee Panda**[1]
**Micah Goldblum**[2]    **Ronen Basri**[3]    **Tom Goldstein**[1]    **David Jacobs**[1]
[1]University of Maryland    [2]Columbia University    [3]Weizmann Institute of Science

## Abstract

Attention is foundational to large language models (LLMs), enabling different heads to have diverse focus on relevant input tokens. However, learned behaviors like attention sinks, where the first token receives the most attention despite limited semantic importance, suggest some heads may be inactive, and point to a significant source of computational redundancy. To analyze this phenomenon, we evaluate 12 score functions that measure different ways a head can be inactive. Thresholding these scores allows us to analyze different sets of potentially inactive attention heads. We evaluate whether identified heads are inactive through model interventions, finding that more than 12% of attention heads are inactive on average, and can be ablated in specific contexts while maintaining MMLU accuracy to within 1% of the pretrained LLM. Across 3 model families, our score functions that measure the average norm of a head's output consistently identify inactive heads that would not have been found by score functions that rely solely on attention weights. We establish that relying on a score function that measures a first token attention sink would underestimate the prevalence of inactive heads, failing to identify more than 7% of inactive heads on average. We also show how measuring score distributions can provide insights into attention behavior. For instance, we find evidence that finetuning causes little to no change in attention behavior, and that even within the same model family, large model scales present different attention behaviors. [1]

## 1 Introduction

Attention is a key component of the transformer architecture, which has led to breakthroughs in language modeling (Radford et al., 2019; Touvron et al., 2023; Dubey et al., 2024; OLMo et al., 2024; Yang et al., 2024). The attention mechanism allows tokens to incorporate information from relevant tokens, with multiple heads of attention capturing different types of relevance (Vaswani, 2017). But several works have found that attention can become "dormant" and concentrate on initial tokens, which are semantically irrelevant (Yu et al., 2024; Chen et al., 2025). The question we seek to answer is: How prevalent are inactive attention heads?

Depending on how one defines the word *inactive*, different answers are possible. Past work has focused exclusively on the attention weights (Guo et al., 2024a), labeling heads as "dormant" using a threshold-based score function. This is motivated by the idea that if a head attends primarily to the first token, and the first token has a near-zero value state, then the head output will be near-zero. However, this reasoning ignores other possibilities. Because an attention head's output is a convex combination of value vectors, a head could attend to multiple tokens with near-zero value states to produce a near-zero output. In this work, we explore multiple ways an attention head can be considered inactive by evaluating a range of score functions and determine the most robust method for identifying inactive heads.

While classifying heads using score functions offers insights into learned attention behaviors, simply identifying potentially inactive heads is not enough. We must also evaluate which set of identified heads best preserves model performance when such heads are zeroed out. Without zeroing out attention head outputs to erase their contribution to the hidden state, it is impossible to verify if identified

---

[1]Code is available at: `https://github.com/psandovalsegura/inactive-heads`

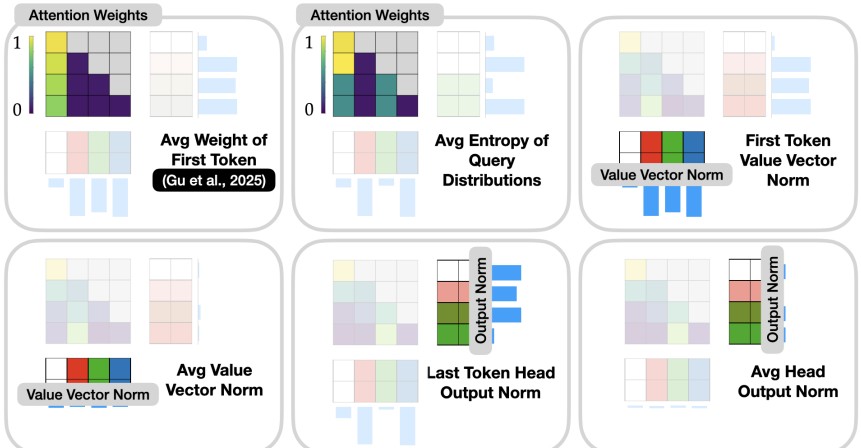

Figure 1: **Score Functions for Inactive Heads.** Our simple score functions measure all three components of attention: attention weights, value vectors, and head output vectors. In each cell, we give an example of the type of attention head that would be identified by each score function once scores are thresholded. For attention weights, the colorbar displays weights. For vectors, color represents direction, and length of the blue bar represents magnitude. For example, the Avg Weight of First Token score (Gu et al., 2025; Guo et al., 2024a) is calculated by computing average weight to the first token, and heads that exhibit high attention to the first token are identified if their their score exceeds a threshold. Value vectors and head outputs do not play a role, which we illustrate with a fade. Including normalized versions of these 6 score functions, there are 12 in total.

heads are actually inactive. To address this, we perform model interventions while evaluating on the MMLU benchmark, and measure the effect of removing identified heads on model accuracy. Using 14 models across 3 model families, this experimental setup also allows us to analyze how attention head behaviors change as models scale and as models are finetuned. We make the following contributions:

- We evaluate 12 score functions that measure distinct properties of attention heads, and we threshold scores to classify attention heads, based on different definitions of *inactive*. We evaluate whether identified heads are truly inactive through model interventions which show that, on average, more than $12\%$ of attention heads are inactive in pretrained LLMs we consider. Using prior characterizations of "dormant heads" would underestimate inactive heads, and fail to identify $7\%$ of heads that are inactive.

- We find that, across model families and across model scale, inactive heads are consistently identified through the same score functions. In particular, measuring the average head output norm is most indicative. By finding a more model-agnostic score function, we provide evidence that inactive heads across transformer LLMs should be understood through head outputs rather than attention weights.

- We demonstrate that analyzing score distributions provides useful insights into attention: they reveal that finetuning induces minimal changes in attention behavior, and that scaling has little effect until models reach large sizes. Together, this suggests attention head behavior is more invariant to common training modifications than one might expect.

While our work has the potential to be used for efficient inference, our focus is strictly on understanding inactive attention heads. Specifically, how can we identify them and evaluate whether they are truly inactive?

## 2 BACKGROUND AND RELATED WORK

The idea that transformers may not be effectively utilizing all attention heads first came up in the context of machine translation. The work of Voita et al. (2019) and Michel et al. (2019) demonstrated

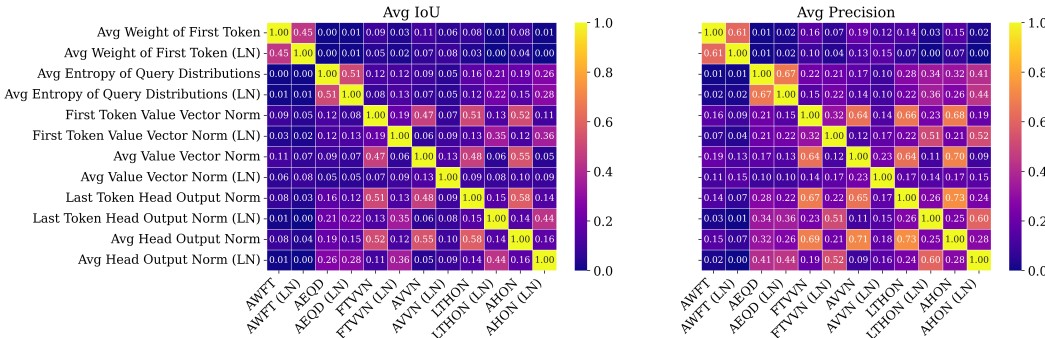

Figure 2: **Score functions identify different sets of heads**. Identifying different sets of heads ensures we capture a broad range of head characteristics, rather than focusing solely on the attention sink pattern of Avg Weight of First Token (Gu et al., 2025). Using 100 FineWeb-Edu training samples, we measure IoU of classifications between each score function on Llama-3.1-8B. We also measure Precision of one score function's classifications relative to another's, using the column score function as ground truth. Column score functions are abbreviated using the first letter of each word, and are in the same order as rows. For each score function, we dynamically choose thresholds such that ~10% of heads are identified as potentially inactive. Even head scores normalized by scores of other heads in the layer, denoted by "(LN)", do not show significant IoU or Precision with their unnormalized counterpart.

that attention heads in transformer models could be removed with minor performance degradations. Methods for determining which heads can be removed have involved optimization of different kinds of objectives based on: stochastic gates (Voita et al., 2019), importance scores (Michel et al., 2019), iterative pruning (Behnke & Heafield, 2020), and subset pruning (Li et al., 2021). In this work, however, we do not use static pruning, where a head is permanently removed. Rather, our model intervention study (Section 4.3) is a form of dynamic pruning, where different heads are dropped at every forward pass. Our objective is to measure how much head computation is wasted in each forward pass by inactive heads that can be zeroed out. We assume inactive heads are those that can be zeroed out, prior to concatenation and output projection of the multi-head attention mechanism.

**Multi-head attention.** The multi-head self-attention mechanism (Vaswani, 2017) allows multiple attention heads to operate in parallel, each focusing on different aspects of the input sequence. For an input of $N$ tokens with dimensionality $d_m$, learned linear projections transform queries, keys, and values ($\mathbf{Q}, \mathbf{K}, \mathbf{V} \in \mathbb{R}^{N \times d_m}$) into lower-dimensional representations: $d_k$-dimensional queries/keys and $d_v$-dimensional values, specific to each of the $h$ heads. The self-attention operation within each head computes attention weights, defined as $\mathbf{A} = \mathrm{softmax}(\frac{\mathbf{Q}\mathbf{K}^\top}{\sqrt{d_k}})$, where $\mathbf{A} \in [0,1]^{N \times N}$ and each row sums to 1, which are then applied to the value vectors. Another way to view the output of the $i^{\text{th}}$ attention head $\mathrm{head}^i \in \mathbb{R}^{N \times d_v}$ is that, for every position in the sequence, we compute a convex combination of value vectors, weighted by the corresponding row of $\mathbf{A}$. The outputs of all heads are concatenated and passed through a final linear projection to produce the module's output. The attention weights, $\mathbf{A}$, are one of the few features we can visualize to get a sense of how pretrained LLMs work. For a token sequence, high attention weights (close to 1) reveal which tokens are considered most relevant. Thus, it is surprising that attention can concentrate on initial tokens, called "attention sinks", as shown in App. Figure 19.

**Attention sinks.** An attention sink or sink token is a token that, despite limited semantic importance, disproportionally receives high attention weight from other tokens in a sequence. They tend to occur either at the first position of the input sequence (Xiao et al., 2024; Guo et al., 2024a), at certain word tokens (e.g., , "and" and "of"), or delimiter tokens (Sun et al., 2024; Yu et al., 2024). The first token can be an attention sink even when it is not a BOS token (Xiao et al., 2024; Gu et al., 2025). To understand the prevalence of attention heads with sinks, Gu et al. (2025) propose to measure the average weight assigned to the first token and check if it exceeds a threshold $\tau$: $\frac{1}{N}\sum_{i=0}^{N-1} \mathbf{A}_{i,0} > \tau$.

Table 1: **Equations of Score Functions.** We define how each score function is used to identify inactive heads with a threshold-based rule. Sample attention heads are in Appendix A.3 Figure 8. We sweep a range of thresholds $\tau$ for each function.

| Score Function | Definition |
|---|---|
| Avg Weight of First Token (AWFT) | $\frac{1}{N} \sum_{i=0}^{N-1} \mathbf{A}_{i,0} > \tau$ |
| Avg Entropy of Query Distributions (AEQD) | $\frac{1}{N} \sum_{i=0}^{N-1} \text{Ent}(\mathbf{A}_{i,:}) < \tau$ |
| First Token Value Vector Norm (FTVVN) | $\|\mathbf{V}_{0,:}\|_2 < \tau$ |
| Avg Value Vector Norm (AVVN) | $\frac{1}{N} \sum_{i=0}^{N-1} \|\mathbf{V}_{i,:}\|_2 < \tau$ |
| Last Token Head Output Norm (LTHON) | $\|\text{head}_{N-1,:}^i\|_2 < \tau$ |
| Avg Head Output Norm (AHON) | $\frac{1}{N} \sum_{i=0}^{N-1} \|\text{head}_{i,:}^i\|_2 < \tau$ |

In our work, we refer to this metric as "Avg Weight of First Token" and use it as a starting point, but ultimately find that different threshold-based score functions are needed for identifying inactive heads in model families outside of Llama (Dubey et al., 2024) and GPT-2 (Radford et al., 2019).

As can be seen in App. Figure 19, attention sinks also exhibit value-state drains (Guo et al., 2024b; Gu et al., 2025; Kobayashi et al., 2020), where the norm of the value state is near zero. Note that if the sink token's value state is zero but the attention weight to the sink token is one, the output of this head (prior to the output projection) is zero, leading Guo et al. (2024a) to call this a dormant attention head.

**Dormant attention.**  Guo et al. (2024a) propose the idea that attention heads can be either active or dormant. They perform a model intervention study on 3 attention heads of Llama-2-7B Base, where a specific attention head output is zeroed out, and the difference in loss as a result of the intervention is measured. They find that the difference in loss can depend on whether input text is from Wikipedia or GitHub; a head that is dormant on Wikipedia samples does not change the loss when the head output is zeroed. In our work, we go beyond analyzing one attention head at a time and propose to zero out all "dormant heads" while evaluating models on a real-world benchmark task. If "dormant heads" are truly inactive, then zeroing them out should have little effect on model performance. We also propose new score functions that measure different definitions of *inactive*. Unlike Guo et al. (2024a) and Gu et al. (2025) who measure only attention weights, our score functions also consider value vectors and head outputs. By considering the range of ways a head can be inactive, we present a more accurate picture of inactive attention heads in pretrained models. A better understanding of inactive attention heads has the potential to be used for efficiency (Li et al., 2021), KV cache reduction (Liu et al., 2023), and compression (Ge et al., 2024).

## 3 EVALUATING SCORE FUNCTIONS FOR INACTIVE HEADS

We outline a number of simple score functions, each capturing different ways a head could be considered inactive. Each score function assigns a score to every attention head in a transformer LLM. By thresholding the scores, we classify and study different sets of heads. We summarize each score function in Table 1. We use the $\ell_2$-norm wherever applicable.

**Attention patterns.**  An attention head could appear inactive due to the patterns in its attention weights. Both overly focused and overly diffuse attention can be considered "inactive," depending on the input. Prior work has recognized the case where attention can concentrate on a single token (Gu et al., 2025; Guo et al., 2024a), where attention is distributed too little, and quantified it by calculating the Average Weight of the First Token (AWFT). However, this does not capture the case where there are multiple sink tokens (See App. Figure 19). To measure when attention concentrates on a few tokens, we choose to measure the Entropy of each Query's Distribution over keys, averaged over all queries. Calculating this amounts to measuring entropy of each row of the attention weights,

then taking an average. If average entropy of the head is low, it implies attention concentrates on few tokens. If a head's average entropy is under a threshold $\tau$, we can classify it as potentially inactive. The smaller we make $\tau$, the more strict we are in classifying heads that exhibit this pattern. Changing the direction of the inequality in Avg Entropy of Query Distributions could capture cases where attention is too uniform (*i.e.*, too much attention to all keys), but we chose not to consider this case because heads in early layers tend to exhibit this pattern, and early layers tend to be important (Sun et al., 2025).

**Value vectors.** An attention head could *appear* active with its attention patterns, but could be inactive if all the value vectors are near-zero. To measure this, we compute Avg Value Vector Norm: the average $\ell_2$-norm of value vectors in the head. It is also possible that while attention patterns may not primarily concentrate on the first token, some tokens could still use the first token as a "value-state drain" (Guo et al., 2024a). To capture heads that do this, we compute the First Token Value Vector Norm, *i.e.*, the $\ell_2$-norm of the value vector corresponding to the first position. As with other score functions, we consider normalizing by the average score of other heads in the layer.

**Head outputs.** Finally, an attention head could be inactive if its output is small. This key assumption is most closely captured by the score functions that measure the head output: when a head produces small outputs, its contribution to the residual stream will be minimal, and these heads can be zeroed with little to no accuracy drop. Because multiple choice (MC) benchmark evaluations retrieve the probability assigned by the model to the correct answer from the probability distribution of the last position, we also propose to measure the Last Token Head Output Norm. To capture cases where most positions have head outputs that are small, we measure the Avg Head Output Norm. More information on why we do not use the circuit-based definition of a head output can be found in Appendix A.1.

**Normalization.** We classify heads using the score functions in Table 1, but we also consider normalization by the score of other heads in the same layer. Abusing terminology, we refer to this as layer normalization, or "(LN)" throughout. We do this because raw scores can vary dramatically across layers and models. For example, in Llama-3.2-3B, a value vector norm of 1.0 may be largest for a head in layer index 0, but smallest in layer index 27. Using a relative score allows us to make meaningful comparisons across layers and models. As an example, the Avg Head Output Norm (AHON) score function for the $i^{\text{th}}$ attention head in a layer is computed as follows: $\text{AvgNorm}(\text{head}^i)$, where $\text{AvgNorm}(T) = \frac{1}{N}\sum_{i=0}^{N}\|T_{i,:}\|_2$ computes the average $\ell_2$-norm of the rows of input matrix $T$ and where $\text{head}^i \in \mathbb{R}^{N \times d_v}$ is the head output. Then, the normalized version, Avg Head Output Norm (LN), is simply normalized by the average AHON score of heads in the same layer:

$$\frac{\text{AvgNorm}(\text{head}^i)}{\frac{1}{N_{\text{layer}}}\sum_{j=0}^{N_{\text{layer}}}\text{AvgNorm}(\text{head}^j)} \tag{1}$$

Thresholding the score of Equation (1) whenever it is $< \tau$ controls how strictly we classify a head as inactive. For example, when $\tau = 0.1$, heads with output norms less than 10% of the layer average are considered inactive. Code implementations are provided in Appendix A.12.

## 4 EXPERIMENTS

First, we study which score function is best at identifying inactive heads. Then, we use head scores to study how attention behaviors change as models scale and are finetuned.

### 4.1 SETUP

We download 14 pretrained models using Hugging Face `transformers` (Wolf et al., 2020): Llama-3.1-8B, Llama-3.1-8B-Instruct (Dubey et al., 2024), Llama-3.2-3B, Llama-3.2-3B-Instruct (Meta, 2024), OLMo-2-1124-7B, OLMo-2-1124-7B-SFT, OLMo-2-1124-7B-DPO, OLMo-2-1124-7B-Instruct (OLMo et al., 2024), Qwen2.5-0.5B, Qwen2.5-1.5B, Qwen2.5-3B, Qwen2.5-7B, Qwen2.5-7B-Instruct, and Qwen2.5-14B (Yang et al., 2024). We refer to models from the same organization as belonging to the same "model family." We evaluate models on MMLU (Hendrycks

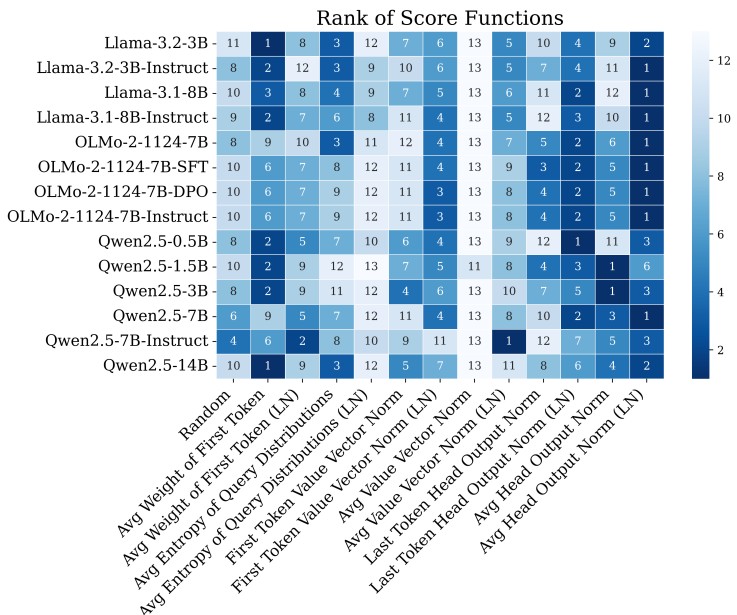

Figure 3: **Measuring Avg Head Output Norm is best at identifying inactive heads for most models**. For every model, we rank the 12 score functions by normalized AUC, which captures their ability to select inactive heads. Avg Head Output Norm (LN) ranks 1st for 8 out of 14 models, and ranks in top-3 for 13 out of 14 models. Top score functions are consistent across model families.

et al., 2021) (5-shot) in our model intervention experiments. For datasets that do not use an LLM-as-a-Judge, MMLU is the multiple-choice (MC) benchmark with the highest Spearman correlation with ChatBot Arena (Li et al., 2023; Chiang et al., 2024), and thus makes it the most appropriate choice for aligning with real-world chatbot preferences while keeping evaluation consistent and tractable. Additional results on PIQA (0-shot) (Bisk et al., 2020) and WinoGrande (5-shot) (Sakaguchi et al., 2021) can be found in Appendix A.4. All evaluations are performed using `lm-evaluation-harness` (Gao et al., 2024). We also use FineWeb-Edu (Lozhkov et al., 2024), which is a pretraining dataset of educational webpages. We randomly truncate FineWeb-Edu training sequences to between 10 and 3000 tokens to exhibit a variety of sequence lengths. Additional model and dataset details are in Appendix A.13.

**Metrics and Thresholds.** In a transformer with $N_{\text{heads}}$ attention heads per $N_{\text{layers}}$ layers, and a dataset $\mathcal{D}$ of token sequences, a forward pass on input sequence $x \in \mathcal{D}$ will produce attention weights, value states, and head outputs for every head. We measure these components, and assign a score to every head, arranged in a matrix $\mathbf{S} \in \mathbb{R}^{N_{\text{heads}} \times N_{\text{layers}}}$. Thresholding $\mathbf{S}$ results in a boolean matrix $\mathbf{B}$ of the same size. In our model intervention experiments, a different boolean matrix is constructed for every new forward pass and "% of Model Heads Zeroed" refers to the percent of `True` values in $\mathbf{B}$, averaged over all token sequences in $\mathcal{D}$. The proportion of heads each score function identifies as inactive can be controlled by the threshold $\tau$, which we vary for each function. The thresholds for each score function are chosen by measuring head scores across MMLU inputs, constructing the CDF of scores, and calculating the $p$-th quantile[2] for $p \in [0, 5, 10, 15, 20, 25, 30]$. This way, we can estimate the proportion of heads that will be selected for each threshold, allowing us to focus on zeroing out at most 30% of heads. All score distributions are in App. Figure 16.

## 4.2 SCORE FUNCTIONS IDENTIFY DIFFERENT SETS OF HEADS

To ensure that each score function measures a different aspect of attention heads, we measure the agreement between predictions (post-thresholding) using IoU. We also measure precision, taking one of the score functions as ground truth. We use Llama-3.1-8B and 100 FineWeb-Edu sequences

---

[2]In the case of Avg Weight of the First Token, we calculate the $(1 - p)$-th quantile because the threshold inequality reversed.

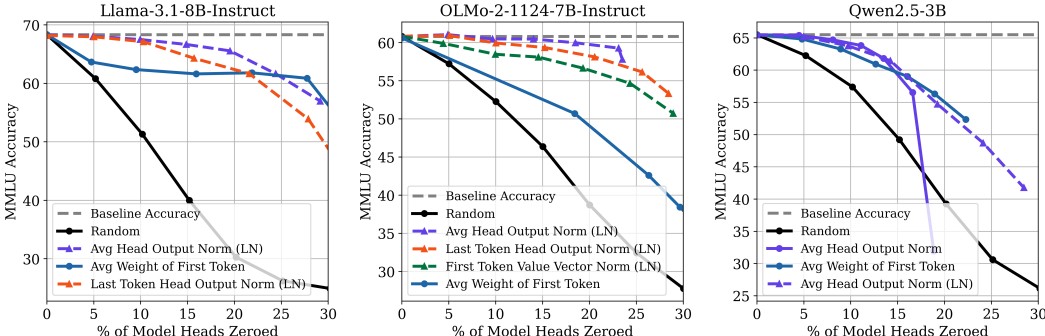

Figure 4: **Inactive heads can be identified and zeroed with minor performance degradation.** For each model, we plot the top-3 score functions. Average Weight of First Token is not in the top-3 for OLMo-2-1124-7B-Instruct, but we include it for comparison to prior work. The gray dotted line represents baseline accuracy of the pretrained model. The black line represents zeroing out heads uniformly at random. The Avg Head Output Norm (LN) score function is best at identifying the most heads, that when zeroed, maintain accuracy. Complete results for all models and score functions can be found in App. Figure 10.

to collect scores, and we threshold them dynamically to make predictions. In Figure 2, we see that the max IoU is $0.58$, while the max Precision is $0.73$. This suggests no score function is effectively the same as another. High agreement tends to occur between unnormalized and normalized pairs of score functions (*i.e.*, AWFT and AWFT (LN)), but not always. For example, Last Token Head Output Norm has an IoU of $0.58$ with Average Head Output Norm. In a similar vein, First Token Value Vector Norm has an IoU of $0.47$ with Average Value Vector Norm. This effect may occur because the presence of small-magnitude vectors drives down the overall average magnitude. The score function that best predicts the same heads as AWFT is AWFT (LN) with a precision of $0.61$. Notably, normalizing by the average score of other heads in the layer significantly reorders heads. All other score functions have Precision under $0.19$ with AWFT. Because each score function identifies a distinct subset of heads, we can verify their predictions knowing that each captures different underlying properties of attention heads.

## 4.3    VERIFYING INACTIVE HEADS THROUGH MODEL INTERVENTIONS

To verify inactive attention heads, we conduct model interventions by zeroing out the outputs of identified heads from each score function, and measuring the resulting impact on model accuracy. For every model, we also compare to a random baseline where attention heads are zeroed uniformly at random. An effective inactive head score function should identify a substantial fraction of heads that can be zeroed out without compromising the performance of the pretrained model. What score function is best at identifying inactive attention heads?

In Figure 4, we measure how MMLU accuracy changes as we zero out heads selected by different score functions. For brevity, we present one model from each model family: Llama-3.1-8B-Instruct, OLMo-2-1124-7B-Instruct, and Qwen2.5-3B. To do this, we vary the threshold of each score function such that identified heads are at most $30\%$ of the model's heads. We plot the top-3 score functions for every model in Figure 4, but complete results for all models and score functions can be found in App. Figure 10. In Table 2, we quantify the percent of heads that can be zeroed while keeping accuracy to within $1\%$ of baseline. We find that, for 8 out of 14 models, the Avg Head Output Norm (LN) score function is best at identifying the most attention heads that when zeroed, maintain accuracy to within $1\%$ of baseline. For 13 out of 14 models, AHON (LN) ranks in the top-3 score functions. Avg Weight of First Token underestimates the number of inactive heads.

The fact that a single, simple threshold-based function like Avg Head Output (LN) can identify the most inactive heads for multiple models, suggests that looking at attention weights is a misleading signal. While heads with attention sink patterns seem to occur in all models we consider (See App Figure 19), our results suggest these patterns are not solely indicative of inactivity. Moreover,

Table 2: **On average, more than 12% of attention heads can be zeroed while keeping average accuracy within 1% of the original model.** For every model, we calculate the highest percentage of heads that can be zeroed using Average Weight of First Token (AWFT), and compare to our score functions (*i.e.*, all score functions excluding AWFT). Higher is better. The overall best score function is also shown. Our score functions only improve or are within less than 0.3% of AWFT. We are able to identify and verify more inactive heads than previously possible with AWFT. Specifically, using AWFT would classify less than 5% of heads as inactive, which misses 7% of heads, on average.

| Model | % of Heads Zeroed | | Best Score Function |
|---|---|---|---|
| | AWFT | Ours | |
| Llama-3.2-3B | 8.34 | 8.05 (-0.29) | Avg Weight of First Token |
| Llama-3.2-3B-Instruct | 6.87 | 13.04 (+6.17) | Avg Head Output Norm (LN) |
| Llama-3.1-8B | 8.56 | 17.11 (+8.55) | Avg Head Output Norm (LN) |
| Llama-3.1-8B-Instruct | 1.01 | 10.97 (+9.95) | Avg Head Output Norm (LN) |
| OLMo-2-1124-7B | 0.42 | 8.34 (+7.93) | Avg Head Output Norm (LN) |
| OLMo-2-1124-7B-SFT | 1.70 | 18.95 (+17.25) | Avg Head Output Norm (LN) |
| OLMo-2-1124-7B-DPO | 2.14 | 20.60 (+18.46) | Avg Head Output Norm (LN) |
| OLMo-2-1124-7B-Instruct | 1.46 | 19.54 (+18.07) | Avg Head Output Norm (LN) |
| Qwen2.5-0.5B | 7.43 | 14.42 (+6.99) | Last Token Head Output Norm (LN) |
| Qwen2.5-1.5B | 7.55 | 7.49 (-0.07) | Avg Weight of First Token |
| Qwen2.5-3B | 5.67 | 8.78 (+3.11) | Avg Head Output Norm |
| Qwen2.5-7B | 1.25 | 7.54 (+6.29) | Avg Head Output Norm (LN) |
| Qwen2.5-7B-Instruct | 1.13 | 5.76 (+4.63) | Avg Value Vector Norm (LN) |
| Qwen2.5-14B | 11.04 | 9.88 (-1.17) | Avg Weight of First Token |
| Average | 4.61 | 12.18 (+7.56) | - |

our results confirm that while attention head patterns may appear "dormant" (Guo et al., 2024a), their outputs are not. Removing attention heads using Avg Weight of First Token as the score is particularly ineffective for OLMo-2 models. While it is appealing to look only at attention patterns as a consistent feature across models of varying organizations and architectures, it appears that small head outputs are consistently more indicative of inactivity.

Despite looking for negligible head outputs, the complexity and non-linearity of LLMs make it difficult to know whether heads with "small" outputs can be zeroed while maintaining accuracy. To the best of our knowledge, we are the first to systematically evaluate the degree of head inactivity and investigate whether these heads are unnecessary. **Takeaway:** Inactive attention heads are best identified by measuring Avg Head Output Norm (LN). Measuring attention weight patterns like Avg Weight of First Token are not model-agnostic.

**Ranking Score Functions.** Each of 12 score functions use a different set of 7 thresholds. Accuracy curves that terminate at different x-axis values make it difficult to assess which method is best. So, we choose to measure performance by Area Under the Curve (AUC) normalized by the x-axis span of each accuracy curve. For every model, the ranking of each score function is displayed in Figure 3. Notably, Avg Head Output Norm and its normalized version are more model-agnostic, ranking 1st for 10 out of 14 models. In contrast, Avg Weight of First Token ranks 1st for only Llama-3.2-3B and Qwen2.5-14B, and performs most poorly for the OLMo-2 family of models.

## 4.4 INACTIVE HEADS BY LAYER

To better understand the poor performance of Average Weight of First Token (AWFT) on the OLMo-2 family, we investigate the stability of two score functions on different data distributions. Using OLMo-2-1124-7B-Instruct, we evaluate 3 datasets: MMLU (Hendrycks et al., 2021), PIQA (Bisk et al., 2020), and WinoGrande (Sakaguchi et al., 2021), and plot the layer-wise percentage of identified inactive heads in Figure 5. We compare Average Weight of First Token (AWFT) (Gu et al., 2025) against Average Head Output Norm (LN), the best performing method in Section 4.3.

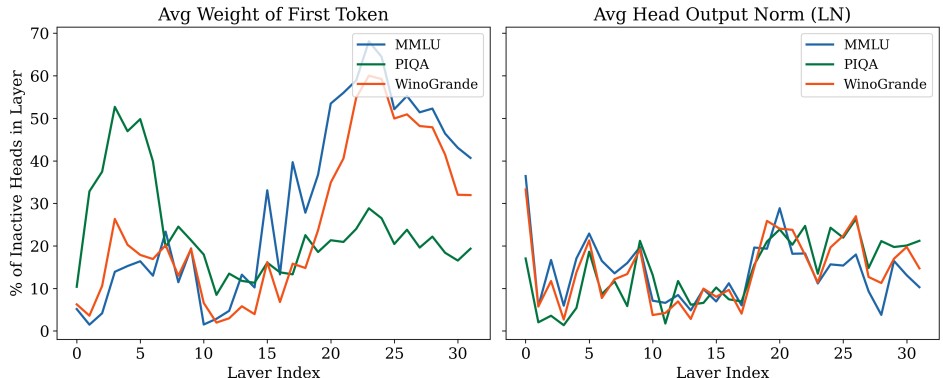

Figure 5: **Average Head Output Norm is a stable measure layer-wise inactivity.** Layer-wise percentage of inactive heads for OLMo-2-1124-7B-Instruct across MMLU, PIQA, and WinoGrande. (Left) Identifying inactive heads via Average Weight to First Token (AWFT) shows high sensitivity to the dataset. (Right) In contrast, using Average Head Output Norm (LN) yields more stable, dataset-agnostic percentages of layer inactivity.

To ensure a fair comparison, we calibrate thresholds for both score functions such that $15\%$ of heads are identified as potentially inactive. Notably, achieving this target classification with AWFT requires highly disparate thresholds depending on the task ($\tau_{\text{MMLU}} = 0.077$, $\tau_{\text{PIQA}} = 0.265$, $\tau_{\text{WinoGrande}} = 0.109$). Conversely, the thresholds required for AHON (LN) are more consistent across datasets ($\tau_{\text{MMLU}} = 0.457$, $\tau_{\text{PIQA}} = 0.435$, $\tau_{\text{WinoGrande}} = 0.473$), indicating a higher degree of generalizability.

The instability of AWFT is further evidenced by the layer-wise percentage of inactive heads. When using AWFT (Figure 5, Left), the per-layer inactive percentages exhibit high sensitivity to the dataset. For instance, on MMLU, inactive heads are concentrated in later layers (peaking above $60\%$ in layer 23). On PIQA, however, large fractions of inactive heads are in early layers (above $50\%$ in layer 3). In contrast, identifying inactive heads via AHON (Figure 5, Right) mitigates this variance and shows stable inactive percentages across all three datasets. Measuring Avg Head Output Norm (LN) provides a dataset-agnostic measure of head inactivity.

## 4.5 Score Distributions for Studying Attention Behaviors

As described in Section 3, each score quantifies a characteristic about an attention head: how much attention is paid to the first token, how small is the first value vector, how small is the average head output, etc. Analyzing the scores of heads can give us a broad idea of attention head behavior. Every model and score function combination produces a distribution of scores. By measuring the similarity of these score distributions, we can determine whether attention behavior is similar or different. With the exception of the AWFT score function, most score distributions have similar modes (See App. Figure 16). Thus, we chose the Wasserstein distance to measure similarity between head score distributions. In Figure 6, we see that AWFT score distributions are similar within a model family, illustrated by a dark block diagonal pattern. The only outlier is Qwen2.5-14B, which has distributions more similar to Llama models than Qwen2.5 models. For AHON (LN) score distributions, Llama and OLMo-2 models are more similar to each other than to Qwen2.5 models.

We use the Qwen2.5 models (Yang et al., 2024) to understand how model scale affects attention head behaviors. There are 5 model types ranging from 0.5B parameters to 14B parameters, but while the models share the same pre-training data, the exact number of training tokens each model was exposed to is not publicly documented. Assuming model scale is the only variable, it appears that model scale impacts attention head behaviors. As Qwen2.5 models grow in size, they are quite similar to one another, until the 14B scale. For example, on most score functions, Wasserstein distance is near-zero, indicated by a clear dark region that corresponds to inter-model similarity. But at the 14B scale, there appears to be only 2 score function distribution where Qwen2.5-14B is similar to other models in the Qwen2.5 family: Avg Value Vector Norm and its layer normalized

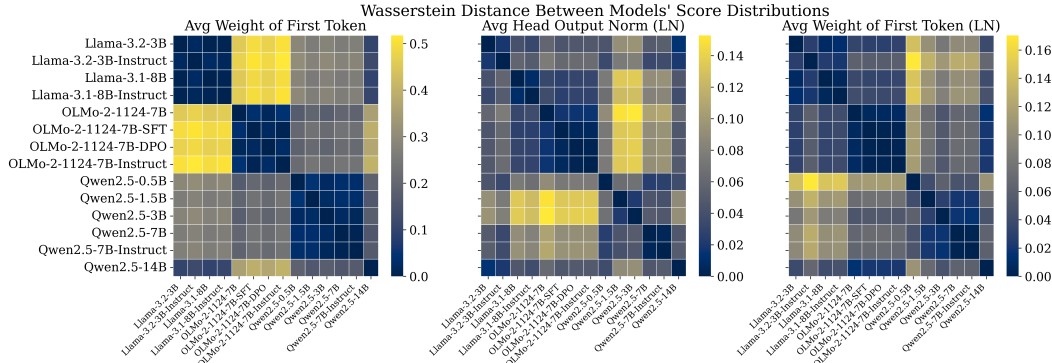

Figure 6: **Head score distributions can be similar within a model family**. Smaller Wasserstein distances indicate similar score distributions. Full results for all 12 score functions can be found in App. Figure 17.

version. This may suggest that larger models learn to specialize their heads in different ways. The similarity of score distributions between the Llama-3 models we consider may have more to do with their training objective as opposed to training data. In particular, because Llama-3.2 was created from pruning Llama-3.1-8B, then performing knowledge distillation, these models are *trained* to be similar (Meta, 2024).

To study finetuning, we primarily rely on the OLMo-2 models which have checkpoints after SFT, after DPO (Rafailov et al., 2023), and after RLHF (Lambert et al., 2024). We also have the Instruct models corresponding to the Llama-3.2-3B, Llama-3.1-8B, and Qwen2.5-7B base models. Figures 6 and 17 illustrate that all finetuning strategies we consider (SFT, DPO, RLHF) have nearly no effect on score distributions, suggesting little change in attention behaviors. Across all score functions, finetuned models have the smallest Wasserstein distances with their base model. In the case of all the OLMo-2 models, the Wasserstein distances of their score distributions are more similar to each other than any other model. These results indicate that finetuning primarily preserves the underlying attention head behaviors of the base model.

## 5 CONCLUSION

Understanding when attention heads are inactive requires looking beyond attention weights. Simple score functions, based on previously ignored components like value vectors and head outputs, can capture a wider array of inactivity. By assigning different scores to every head in a transformer LLM, we can classify different subsets of attention heads as inactive using a threshold. While there is some overlap in the subsets of inactive heads we classify using different score functions, we find that most score functions capture diverse characteristics. By zeroing out identified inactive heads during MMLU evaluations, we find that more than $12\%$ of heads can be zeroed out while maintaining accuracy to within $1\%$ of the baseline model, on average. Using a score function based on a first token attention sink would classify less than $5\%$ of heads as inactive on average, which underestimates the true prevalence of inactive heads. For the majority of models we consider, the score function that measures average head output norm produces the best signal for identifying inactive heads across model families. While attention sinks and sink tokens are almost synonymous with an inactive head, our work demonstrates that there are other sets of heads that are more inactive, and that we can identify them in a more model-agnostic way. Additionally, we show how measuring score distributions can be informative: we use them to explore how finetuning causes little to no change to attention behavior, and how model scale does very little to attention behavior until very large scale. Future work could consider how the MLP module, which proceeds the attention module, could be inactive per-token if converged to an optimal hidden state.

## ACKNOWLEDGMENTS

This work was made possible by National Science Foundation (NSF) grant #2213335. Pedro is supported by a National Defense Science and Engineering Graduate Fellowship (NDSEG) and ac-

knowledges the support and encouragement of John D. Moriarty. RB was supported in part by the Israeli Council for Higher Education (CHE) via the Weizmann Data Science Research Center, by the MBZUAI-WIS Joint Program for Artificial Intelligence Research, and by research grants from the Estates of Tully and Michele Plesser and the Anita James Rosen and Harry Schutzman Foundations. TG was supported in part by the NSF TRAILS Institute (2229885), DARPA TIAMAT, and a grant from Coefficient Giving.

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

## A APPENDIX

### A.1 DEFINING AN ATTENTION HEAD OUTPUT

Vaswani (2017) define an attention head output as the convex combination of value states, resulting in a $d_v$-dimensional vector where $d_v = \frac{d_{\text{model}}}{h}$, $d_{\text{model}}$ is the hidden size, and $h$ is the number of heads. Other works multiply the $d_v$-dimensional vector by a $\mathbb{R}^{d_v \times d_{\text{model}}}$ slice of the output projection matrix (Elhage et al., 2021). This is often called the "circuit-based" definition. In the following, we explain why this choice does not affect our intervention results.

Given $N$ input tokens, an attention head will multiply attention weights $A_i \in \mathbb{R}^{N \times N}$ with value states $V_i \in \mathbb{R}^{N \times d_v}$ so that $Z_i = A_i V_i \in \mathbb{R}^{N \times d_k}$ for $i = 1 \ldots h$, where $i$ is the head index. In multi-head attention, we first concatenate $H = [Z_1; \ldots; Z_h] \in \mathbb{R}^{N \times (h d_v)}$. Then, do an output projection

$$Y = H W_O \quad \text{with} \quad W_O \in \mathbb{R}^{(h d_v) \times d_{\text{model}}}$$

Elhage et al. (2021) partition $W_O$ row-wise into $h$ blocks of size $d_v \times d_{\text{model}}$:

$$W_O = \begin{bmatrix} W_O^{(1)} \\ W_O^{(2)} \\ \vdots \\ W_O^{(h)} \end{bmatrix}, \quad W_O^{(i)} \in \mathbb{R}^{d_v \times d_{\text{model}}} \tag{2}$$

Then they note that the output can be written exactly as the sum of $h$ terms:

$$Y = \sum_{i=1}^{h} Z_i W_O^{(i)} \tag{3}$$

Elhage et al. (2021) use the above to argue that attention heads are independent saying that this is "equivalent to running heads independently, multiplying each by its own output matrix, and adding them into the residual stream." But crucially, before we add the output of the attention module to the residual stream, we actually add up all head contributions (*i.e.*, the summation in Equation (3)). This means that $W_O$ can implement cancellations (if, for example, two head contributions sum to zero) or other special linear combinations of heads.

**Our choice and why it does not affect results in Section 4.3.** Because $W_O$ mixes all heads' outputs into the shared model space, we did not want to integrate these parameters into our measurement. Thus, we define the pre-output-projection $Z_i$ as the attention head output. Importantly, whether we choose to define $Z_i$ or $Z_i W_O^{(i)}$ as the $i^{\text{th}}$ head's output, when "zero out a head" $Z_i$, both will be zero.

Table 3: **Circuit-based definition of head output.** We measure the percent of model heads that can be removed while maintaining accuracy within 1% of baseline (higher is better). For 3 different models evaluated on MMLU, using the circuit-based definition of a head output is comparable to using the original definition.

| Model | AWFT | AHON (LN) | Circuit-based AHON (LN) |
|---|---|---|---|
| Llama-3.1-8B-Instruct | 1.01 | **10.97** | 9.88 |
| Qwen2.5-3B | 5.67 | **7.29** | 5.95 |
| Qwen2.5-7B | 1.25 | 7.54 | **9.04** |
| Average | 2.64 | **8.60** | 8.29 |

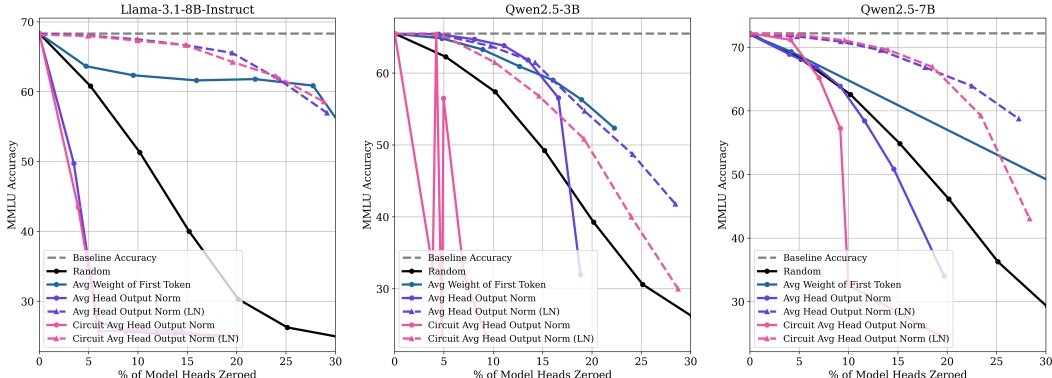

Figure 7: **Circuit-based definition of head output**. For 3 different models evaluated on MMLU, using the normalized circuit-based definition of a head output is comparable to using the original definition. Dashed lines denote layer normalized (LN) score functions.

**Implementing the circuit-based definition.** We implement the circuit-based definition of Average Head Output Norm (LN), and evaluate it on MMLU using 3 models in Figure 7. More specifically, this means that for every head of index $i$ we measure the norms of the rows of $Z_i W_O^{(i)} \in \mathbb{R}^{N \times d_{\text{model}}}$, then average them to produce a score. Table 3 quantitatively shows that the percent of heads that can be zeroed using both definitions is comparable.

## A.2 ADDITIONAL NORMALIZATION STRATEGIES

In Section 3, we describe normalization by the score of other heads in the same layer. For score functions that measure a particular token in the sequence, like Last Token Head Output Norm, it is also possible to normalize by other vectors within the same head. We refer to normalization by other vectors in the same head as head normalization, or "(HN)" within this Appendix. We explored this normalization strategy with the Last Token Head Output Norm score function, but found it was always worse than the layer normalization, "(LN)". HN is only included in App. Figures 10 to 12. In this Appendix, when we refer to 13 score functions, we mean the 12 score function from the main body plus Last Token Head Output Norm (HN).

## A.3 SAMPLE INACTIVE HEADS IDENTIFIED BY SCORE FUNCTIONS

In Figure 8, we give examples of attention heads that are identified by each unnormalized score function, by using threshold-based rules of Table 1. To keep each attention weights matrix small, we use a random, short 50 token sequence from FineWeb-Edu and execute a forward pass on Qwen2.5-7B and OLMo-2-1124-7B. In Figure 9, we show how the normalized AHON (LN) score function can capture distinct kinds of heads. For example, the first head has a first token sink, the second head has two sinks, and the third head has a second token sink. AHON (LN) is able to identify all three as inactive.

As a summary of the kind of head that each score function is meant to capture, we provide a short explanation for each:

- Avg Weight of First Token (AWFT): When attention concentrates on first token
- Avg Entropy of Query Distributions (AEQD): When attention concentrates on a few tokens
- First Token Value Vector Norm (FTVVN): When first value vector is near-zero
- Avg Value Vector Norm (AVVN): When most value vectors are near-zero
- Last Token Head Output Norm (LTHON): When last head output token near-zero
- Avg Head Output Norm (AHON): When most head output tokens are near-zero

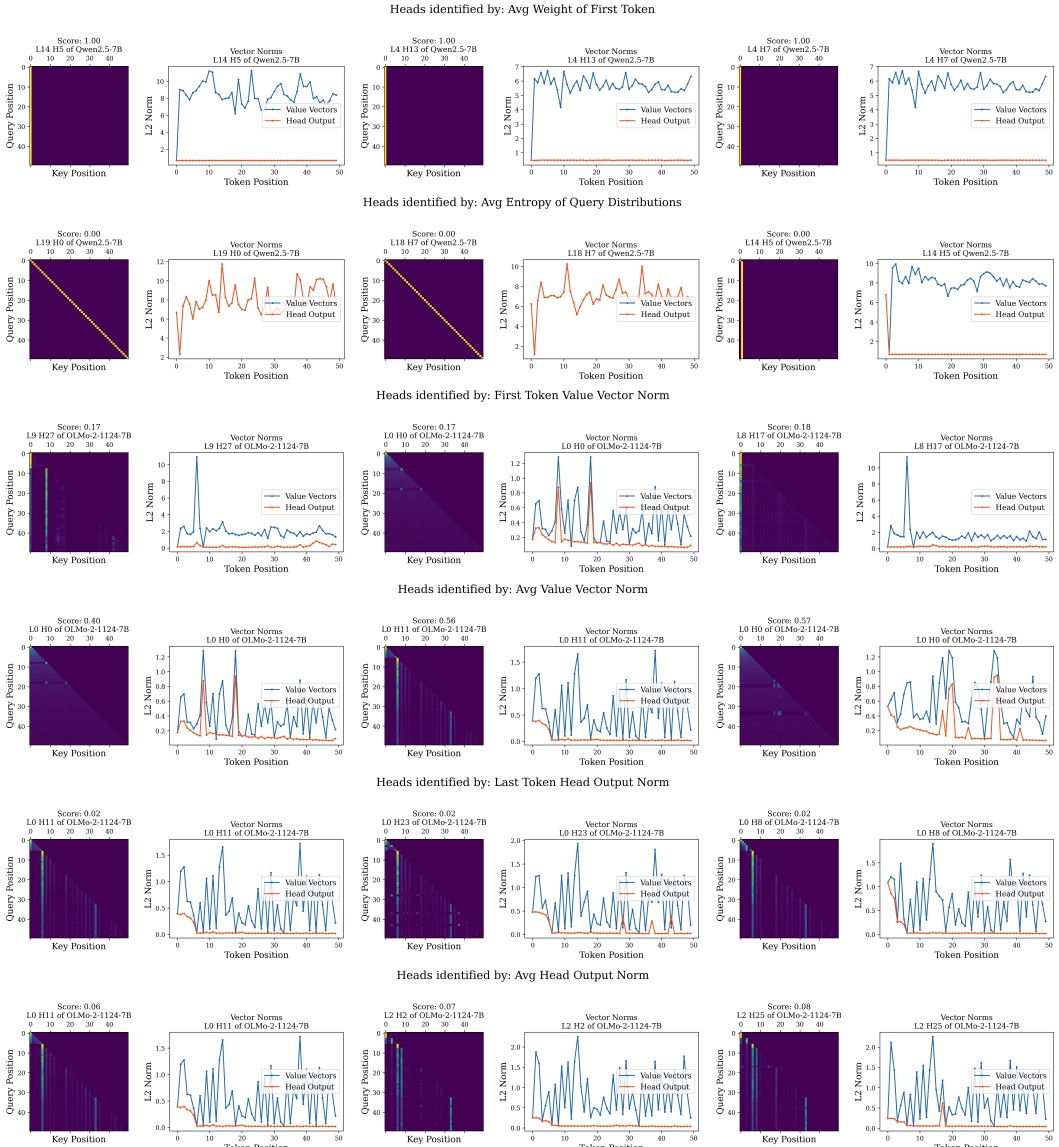

Figure 8: **Sample Attention Heads for 6 Unnormalized Score Functions.** In each row, we plot 3 sample attention heads for each score function in Table 1. To the right of each attention weights matrix, we plot the $\ell_2$-norm of Value Vectors and Head Outputs. We also include the layer index and head index of each attention head above each plot. Lower scores are better for all score functions other than AWFT. Note the variety of heads that are identified by each score function.

## A.4 FULL RESULTS OF SECTION 4.3

In Section 4.3 Figure 4, we present intervention results for 3 models. Results for all 14 models are presented in Figure 10.

### A.4.1 ADDITIONAL DATASETS

We also run the same experiments and analysis on the PIQA (0-shot) (Bisk et al., 2020) and Wino-Grande (5-shot) Sakaguchi et al. (2021) datasets. Plots on how accuracy degrades with different percent of model heads zeroed can be found in Figures 11 and 12. The quantitative analysis, where we measure the maximum proportion of heads that we can zero while maintaining accuracy to within

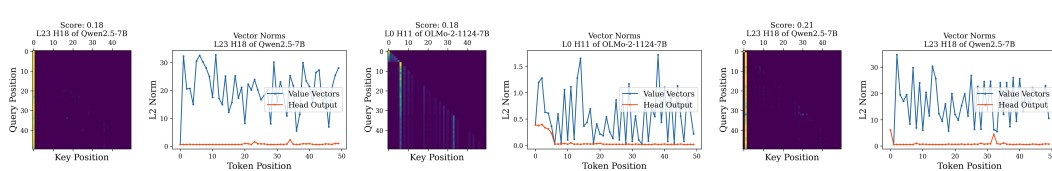

Figure 9: **Sample Attention Heads using Avg Head Output Norm (LN).** We plot 3 sample attention heads for AHON (LN), the best score function by AUC (See Figure 3). To the right of each attention weights matrix, we plot the $\ell_2$-norm of Value Vectors and Head Outputs. Note that the first head has a first token sink, the second head has two sinks, and the third head has a second token sink. AHON (LN) is able to capture all three distinct heads.

1% of the original model can be found in Tables 4 and 5. On PIQA, our score functions show that more than 21% of attention heads can be zeroed while keeping average accuracy within 1% of the original model. On WinoGrande, more than 14% of attention heads can be zeroed while keeping average accuracy within 1% of the original model. Thus, our claim in the main body of the paper is a conservative estimate of inactive heads. In all cases, our score functions can identify more heads than Avg Weight of First Token (AWFT).

Table 4: **PIQA Benchmark: On average, more than 21% of attention heads can be zeroed while keeping average accuracy within 1% of the original model.** For every model, we calculate the highest percentage of heads that can be zeroed using Average Weight of First Token (AWFT), and compare to our score functions (*i.e.*, all score functions excluding AWFT). Higher is better. The overall best score function is also shown. We are able to identify and verify more inactive heads than previously possible with AWFT.

| Model | % of Heads Zeroed | | Best Score Function |
|---|---|---|---|
| | AWFT | Ours | |
| Llama-3.2-3B | 21.76 | 17.37 (-4.39) | Avg Weight of First Token |
| Llama-3.2-3B-Instruct | 15.32 | 25.78 (+10.46) | Avg Head Output Norm (LN) |
| Llama-3.1-8B | 0.00 | 17.85 (+17.85) | First Token Value Vector Norm (LN) |
| Llama-3.1-8B-Instruct | 31.30 | 25.39 (-5.91) | Avg Weight of First Token |
| OLMo-2-1124-7B | 20.74 | 22.70 (+1.97) | Avg Head Output Norm (LN) |
| OLMo-2-1124-7B-SFT | 1.87 | 20.12 (+18.26) | Last Token Head Output Norm (LN) |
| OLMo-2-1124-7B-DPO | 2.93 | 15.87 (+12.94) | First Token Value Vector Norm (LN) |
| OLMo-2-1124-7B-Instruct | 2.54 | 26.83 (+24.30) | Avg Head Output Norm (LN) |
| Qwen2.5-0.5B | 0.00 | 18.36 (+18.36) | Avg Head Output Norm (LN) |
| Qwen2.5-1.5B | 24.33 | 21.32 (-3.01) | Avg Weight of First Token |
| Qwen2.5-3B | 16.51 | 21.42 (+4.91) | Avg Head Output Norm |
| Qwen2.5-7B | 0.00 | 28.66 (+28.66) | First Token Value Vector Norm |
| Qwen2.5-7B-Instruct | 2.84 | 11.45 (+8.61) | Avg Head Output Norm (LN) |
| Qwen2.5-14B | 0.00 | 26.12 (+26.12) | First Token Value Vector Norm (LN) |
| Average | 10.01 | 21.38 (+11.37) | - |

All models and score function evaluations can be found in Figure 10.

## A.5 MULTIPLE FORWARD PASS ANALYSIS

The main body of the paper focuses on analyzing a single forward pass because that is how log-likelihood evaluations of MMLU, PIQA, WinoGrande and other MC benchmarks are performed. To explore how heads may change from inactive to active, based on increasing context, we choose 3 FineWeb-Edu train sequences of at least 512 OLMo-2 tokens in length:

Table 5: **WinoGrande Benchmark: On average, more than 14% of attention heads can be zeroed while keeping average accuracy within 1% of the original model.** For every model, we calculate the highest percentage of heads that can be zeroed using Average Weight of First Token (AWFT), and compare to our score functions (*i.e.*, all score functions excluding AWFT). Higher is better. The overall best score function is also shown. We are able to identify and verify more inactive heads than previously possible with AWFT.

| Model | % of Heads Zeroed | | Best Score Function |
|---|---|---|---|
| | AWFT | Ours | |
| Llama-3.2-3B | 9.11 | 11.64 (+2.53) | Avg Head Output Norm (LN) |
| Llama-3.2-3B-Instruct | 5.57 | 20.37 (+14.79) | Avg Head Output Norm (LN) |
| Llama-3.1-8B | 6.86 | 17.08 (+10.22) | Avg Head Output Norm (LN) |
| Llama-3.1-8B-Instruct | 12.44 | 11.89 (-0.55) | Avg Weight of First Token |
| OLMo-2-1124-7B | 15.39 | 16.12 (+0.73) | Avg Head Output Norm (LN) |
| OLMo-2-1124-7B-SFT | 2.32 | 19.00 (+16.68) | Avg Head Output Norm (LN) |
| OLMo-2-1124-7B-DPO | 2.79 | 16.94 (+14.15) | Avg Head Output Norm (LN) |
| OLMo-2-1124-7B-Instruct | 3.02 | 20.91 (+17.89) | Last Token Head Output Norm (LN) |
| Qwen2.5-0.5B | 6.49 | 11.35 (+4.86) | Avg Head Output Norm (LN) |
| Qwen2.5-1.5B | 7.49 | 12.68 (+5.18) | First Token Value Vector Norm |
| Qwen2.5-3B | 5.73 | 10.32 (+4.60) | First Token Value Vector Norm |
| Qwen2.5-7B | 6.60 | 15.69 (+9.10) | First Token Value Vector Norm (LN) |
| Qwen2.5-7B-Instruct | 2.00 | 8.03 (+6.03) | Avg Value Vector Norm (LN) |
| Qwen2.5-14B | 10.80 | 10.25 (-0.55) | Avg Weight of First Token |
| Average | 6.90 | 14.45 (+7.55) | - |

- Seq 0 ID: urn:uuid:0d8a309d-25c5-405d-a08a-c11239f0d717. Category: Literary commentary.

- Seq 1 ID: urn:uuid:316c7af5-14e1-4d0b-9576-753e17ef2cc5. Category: Popular Science or Journalism.

- Seq 2 ID: urn:uuid:c337bcd8-6aa1-4f2d-8c48-b916442ebbee. Category: Educational article on software licensing.

We choose to evaluate inactive heads of OLMo-2-1124-7B-Instruct as each of these sequences grows in Figure 13. We analyze AWFT and AHON (LN) score functions because AWFT is prior work, and AHON (LN) is the best score function we discover in Section 4.3. As expected, AWFT presents decaying curves where the model has fewer inactive heads as sequence length increases. As sequences increase in length, each query has more potential tokens it can attend to, making it less likely that the first token receives disproportionate attention. Additionally, as can be seen in Figure 20 and Appendix A.3, OLMo-2 models tend to use additional sink tokens (not just the first token). In contrast, measuring AHON (LN) inactive heads as sequence length increases, we observe noisy measurements of inactive head proportions in short sequences that converge as each sequence gets sufficiently long. In measuring both AWFT and AHON (LN) inactive heads, it is clear that inactivity is context-dependent.

We are also interested in where changing attention heads are located. We examine inactive head masks for AWFT and AHON (LN) in Figures 14 and 15. Every matrix cell indicates a particular attention head, and we color yellow any head that is identified as inactive. Unlike AWFT score function measurements, AHON (LN) inactive heads are more spread out throughout the model. We also observe the same phenomenon in Section 4.4 Figure 5.

## A.6 SCORE DISTRIBUTIONS ON MMLU

All score function distributions, from MMLU, can be found in Figure 16.

## A.7 FULL RESULTS OF SECTION 4.5 WASSERSTEIN DISTANCES

All Wasserstein distance matrices, for all score functions, can be found in Figure 17.

## A.8 SCORE DISTRIBUTIONS COMPARISON: FINEWEB-EDU VS. MMLU

In Figure 18, we show how the distribution of Avg Weight of First Token scores change when the input dataset changes. Qwen2.5-7B has a small fraction of heads that exceed $0.8$ average attention to the first token on FineWeb-Edu, but on MMLU there are essentially none. One explanation is the token sequence length which, for 5-shot MMLU, tend to be approximately 3000 tokens long. In contrast, the FineWeb-Edu sequences we use are randomly truncated to be between 30 and 3000 tokens.

## A.9 ATTENTION SINKS AND VALUE-STATE DRAINS

Sample attention sink patterns and value-state drains can be found in Figure 19.

## A.10 REDUNDANCY OF ATTENTION PATTERNS

Numerous works have noted the prevalence of redundant attention patterns across attention heads, even from different layers Xiao et al. (2024); Mu et al. (2024); Guo et al. (2024a); Liu et al. (2023); Ge et al. (2024), and attention sinks make up part of those patterns. In Figure 20, we observe the same phenomena in recent pretrained LLMs. We plot attention matrices from the last 8 layers of five models. The chosen head indices are simply ordered sequentially. Not only are attention patterns similar among heads of every model, different models have different sink token behaviors. Llama-2-7b-hf uses two sinks and divides weight between them. Llama-3.1-8B uses the first token as a sink. OLMo-2-1124-7B, like Llama-2-7b-hf, uses two sink tokens, but the second intermediate sink is at a different position than that of Llama-2-7b-hf. Qwen2.5-7B tends to use the *second* token as a sink, while Qwen2.5-14B uses the first token. We exclude Llama-3.2-3B from Figure 20 because it presents similar attention patterns to Llama-3.1-8B.

For every model in Figure 20, we also show the top 2 principal components across all attention matrices (of which there are $N_{\text{layer}} \times N_{\text{heads}}$). It is surprising to see a few principal components capturing more than half of the observed variance of attention matrices, for all models.

The input to all models in Figure 20 is from MMLU's `high_school_computer_science`, dev split, index 2. The input text is in the same format used by `lm-evaluation-harness` (Gao et al., 2024):

```
The following are multiple choice questions (with answers) about
high school computer science.

What is the output of "abc"[::-1] in Python 3?
A. Error
B. abc
C. cba
D. c
Answer:
```

## A.11 ON LM-EVALUATION-HARNESS EVALUATIONS

For reliable and complete evaluations, use batch_size='auto' in lm-evaluation-harness. It not only finds an optimal batch size for your hardware to prevent memory errors but also mitigates the issue where an entire batch may be skipped because one sample cannot fit on the GPU.

## A.12 PYTORCH IMPLEMENTATION OF AVG HEAD OUTPUT NORM (LN) AND AVG WEIGHT TO FIRST TOKEN

We provide sample PyTorch code (Paszke, 2019), for Avg Head Output Norm (LN) and Avg Weight to First Token, that can be integrated into a self-attention module's forward pass. When using

`lm-evaluation-harness` (Gao et al., 2024) to evaluate pretrained models, additional steps must be taken to ignore padding tokens during log likelihood evaluations (on MC datasets).

All score functions are implemented within the self-attention module and take as input the following tensors: attention weights of size $(B, N_{\text{head}}, S, S)$, value states of size $(B, N_{\text{head}}, S, d_v)$, and a float threshold. $B$ denotes the batch size, $S$ denotes the sequence length, $N_{\text{head}}$ denotes the number of attention heads, and $d_v$ is the dimension of the value states. As output, both implementations return a boolean mask of dormant heads of size $(B, N_{\text{head}})$ and attention outputs of size $(B, N_{\text{head}}, S, d_v)$. In the dormant mask, an entry is True if the head is declared dormant, and False otherwise. Note that models that use grouped-query attention (GQA) (Ainslie et al., 2023) or multi-query attention (MQA) (Shazeer, 2019) still construct tensors of these sizes, so the specific kind of multi-head attention is not relevant.

Listing 1: Avg Weight to First Token in PyTorch

```
def awft_dormant_mask(attn_weights, value_states, threshold):
    avg_weight = attn_weights.mean(dim=-2)          # (B, N_head, S)
    first_token_avg_weight = avg_weight[:,:,0]       # (B, N_head)
    dormant_mask = first_token_avg_weight > threshold # (B, N_head)

    # Model intervention: set dormant head outputs to zero
    attn_output = torch.matmul(attn_weights, value_states)
    attn_output[dormant_mask] = 0
    return attn_output, dormant_mask
```

Listing 2: Avg Head Output Norm (LN) in PyTorch, following Equation (1)

```
def ahon_ln_dormant_mask(attn_weights, value_states, threshold):
    attn_output = torch.matmul(attn_weights, value_states)
    norm_per_token = attn_output.norm(dim=-1)        # (B, N_head, S)
    avg_norm_per_head = norm_per_token.mean(dim=-1)  # (B, N_head)

    # compute average across all heads in layer
    layer_context = avg_norm_per_head.mean(dim=1)    # (B,)
    rel_avg_norm_per_head = (avg_norm_per_head / layer_context[:, None])
        # (B, N_head)
    dormant_mask = rel_avg_norm_per_head < threshold # (B, N_head)

    # Model intervention: set dormant head outputs to zero
    attn_output[dormant_mask] = 0
    return attn_output, dormant_mask
```

### A.13 ADDITIONAL MODEL AND DATASET INFORMATION

**Model details.** Model inference is done in the original data type of the saved weights using `AutoModelForCausalLM.from_pretrained(...,torch_dtype="auto")`, except for OLMo-2 models, where we use float16. All models are downloaded using Hugging Face (HF) `transformers` (Wolf et al., 2020). Parameter counts and release dates are shown in Table 6.

| Model Name | Params | Heads Per Layer | Num Layers | Total Heads | Release Date |
|---|---|---|---|---|---|
| Llama-3.1 8B | 8.03B | 32 | 32 | 1024 | Jul 2024 |
| Llama-3.2-3B | 3.21B | 24 | 28 | 672 | Sep 2024 |
| OLMo-2-1124-7B | 7.3B | 32 | 32 | 1024 | Nov 2024 |
| Qwen2.5-7B | 7.62B | 28 | 28 | 784 | Feb 2025 |
| Qwen2.5-14B | 14.8B | 40 | 48 | 1920 | Feb 2025 |

Table 6: Additional model information of pretrained LLMs used in this work.

**Dataset details.** The test split of the MMLU benchmark contains 14,042 multiple-choice (MC) questions spanning 57 tasks including mathematics, US history, computer science, law, and more. Each question has 4 answer choices.

We also use PIQA (Bisk et al., 2020) and WinoGrande (Sakaguchi et al., 2021), but only in Appendix A.4 to reduce clutter in the main body. PIQA is a MC benchmark dataset of physical commonsense questions where each question has 2 answer choices. WinoGrande is a MC benchmark dataset of pronoun resolution problems where each problem has 2 answer choices.

### A.14 PRACTICAL IMPLICATIONS AND LIMITATIONS

**Architectures.** We think there is evidence that different problems require different fractions of model parameters to solve (Raposo et al., 2024; Lin et al., 2017). Our experiments provide concrete evidence that this occurs in self-attention, and we think sparsity can be beneficial for designing architectures like MoEs but for attention modules. We may not need to have all heads in memory if only a subset will be used. A router could potentially be developed to select the appropriate heads. Our work provides necessary background on inactive attention heads so that future work can develop new architectures for this problem.

**Efficiency.** It should be possible to dynamically prune different heads on-the-fly whenever we identify a head as inactive. But it is unknown whether the overhead in identifying these heads (by computing scores) will make inference so slow that the compute savings are not worth it. In particular, our best score function (AHON (LN)) has the issue that it requires computing the head output itself, making it impossible to save compute when "skipping" a head. However, future work may discover a more proper score function that would allow for compute saving. For example, Lin et al. (2017) showed that image classifiers could be dynamically pruned at runtime, for each individual input. Another potential direction is KV cache compression: specifically, not storing K/V for heads that are inactive. Our work also opens up new directions for investigation like finetuning inactive heads.

**Limitations.** We focus on analyzing a single forward pass because that is how log-likelihood evaluations of MMLU, PIQA, WinoGrande and other MC benchmarks are performed. Analyzing generation is complex because zeroing out a single incorrectly identified head can ruin the rest of the generated sequence (*i.e.*, more opportunities for failure). For example, if we zero out too many heads, we could sample a token representing a different language. Analyzing generation is a natural next step, as it is used for more practical use-cases.

### A.15 LLM USAGE

LLMs were used to aid or polish writing. Gemini 2.5 Pro and ChatGPT were used with prompts like "revise this sentence".

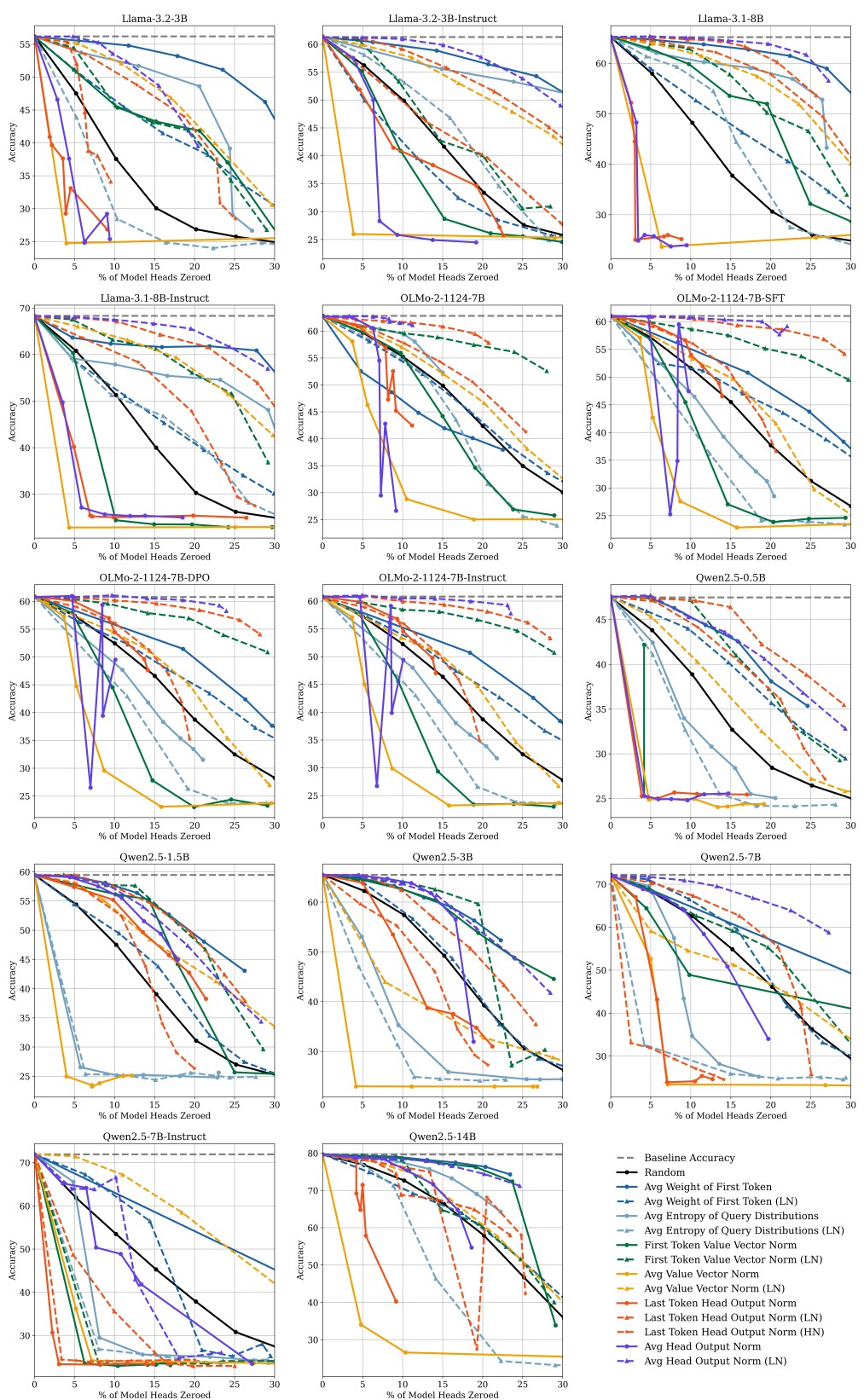

Figure 10: **Complete MMLU Results of Section 4.3.** For 14 models, we consider 13 score functions, and evaluate each at 7 distinct thresholds $\tau$ for each.

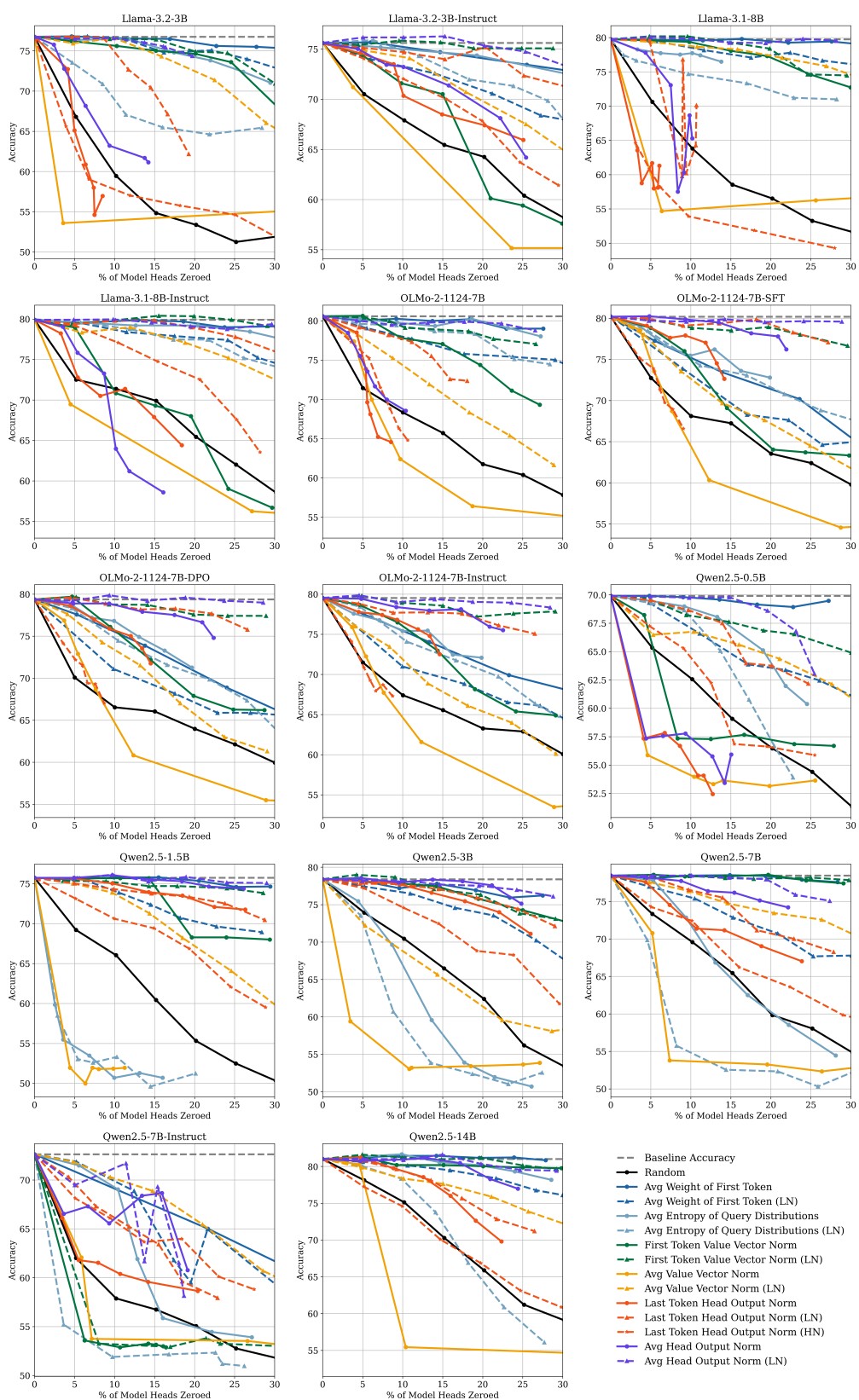

Figure 11: **Additional PIQA Results for Section 4.3.** For 14 models, we consider 13 score functions, and evaluate each at 7 distinct thresholds $\tau$ for each.

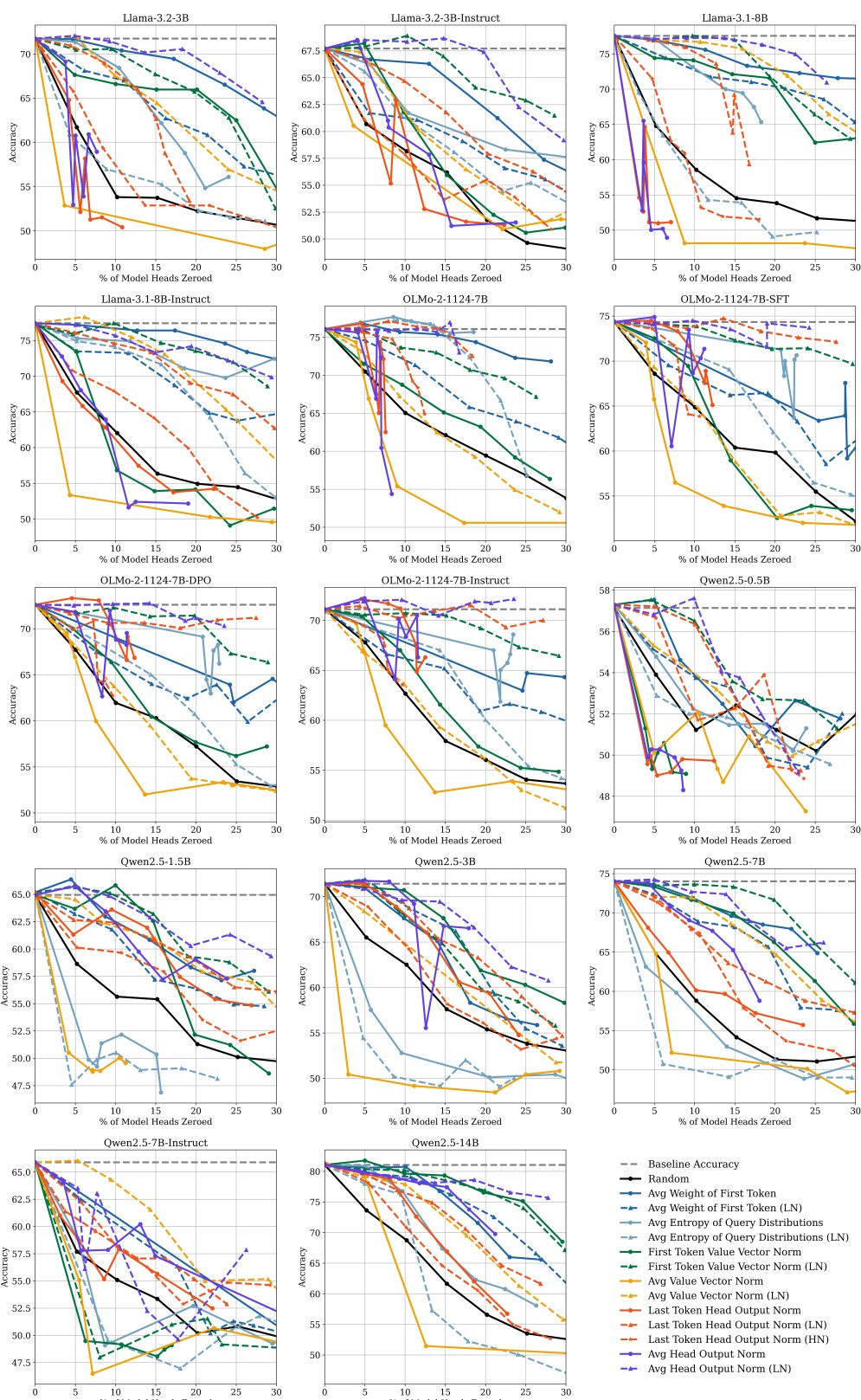

Figure 12: **Additional WinoGrande Results for Section 4.3.** For 14 models, we consider 13 score functions, and evaluate each at 7 distinct thresholds $\tau$ for each.

Model: OLMo-2-1124-7B-Instruct

Figure 13: **Inactive Heads as Sequence Length Changes.** For OLMo-2-1124-7B-Instruct, we consider 3 FineWeb-Edu sequences, and evaluate the percent of inactive heads in the model as the sequence length increases. We compare AWFT ($\tau = 0.077$) and AHON (LN) ($\tau = 0.457$) score functions.

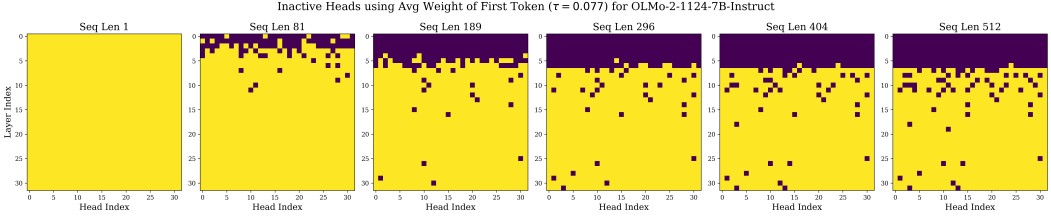

Figure 14: **Inactive Heads as Sequence Length Changes.** For OLMo-2-1124-7B-Instruct, we plot an inactive head mask for increasing sequence lengths of "Seq 0" from Figure 13. Yellow indicates an inactive head, as identified by AWFT score function.

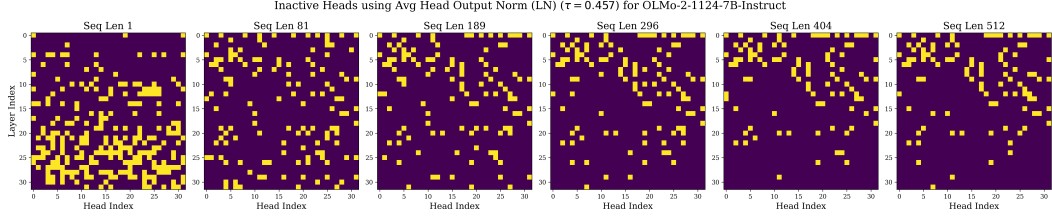

Figure 15: **Inactive Heads as Sequence Length Changes.** For OLMo-2-1124-7B-Instruct, we plot an inactive head mask for increasing sequence lengths of "Seq 0" from Figure 13. Yellow indicates an inactive head, as identified by AHON (LN) score function.

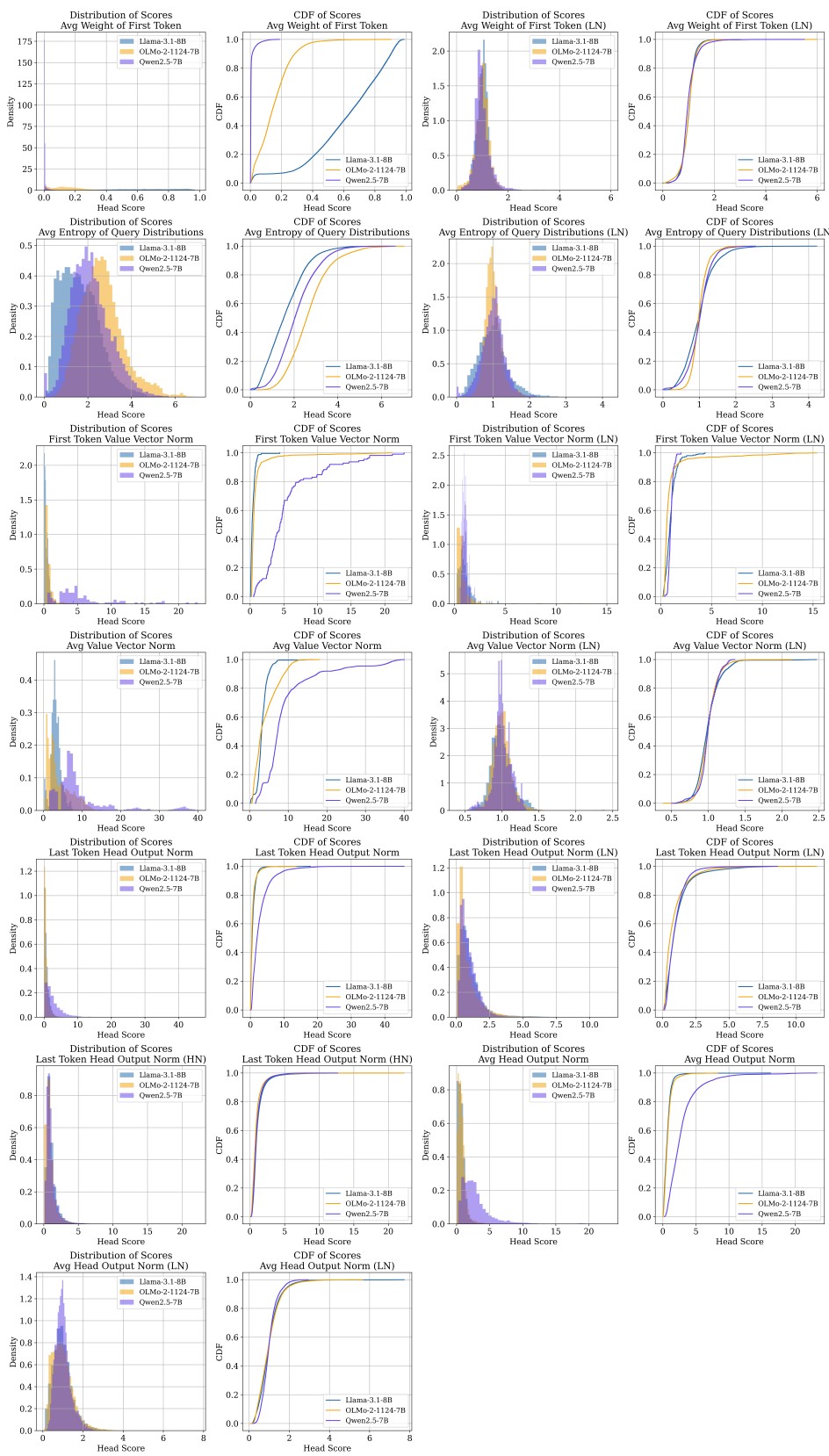

Figure 16: **Head Score Distributions on MMLU.** For 13 score functions, we plot a density distribution and a CDF of head scores. We only show scores for Llama-3.1-8B, OLMo-2-1124-7B, and Qwen2.5-7B for brevity.

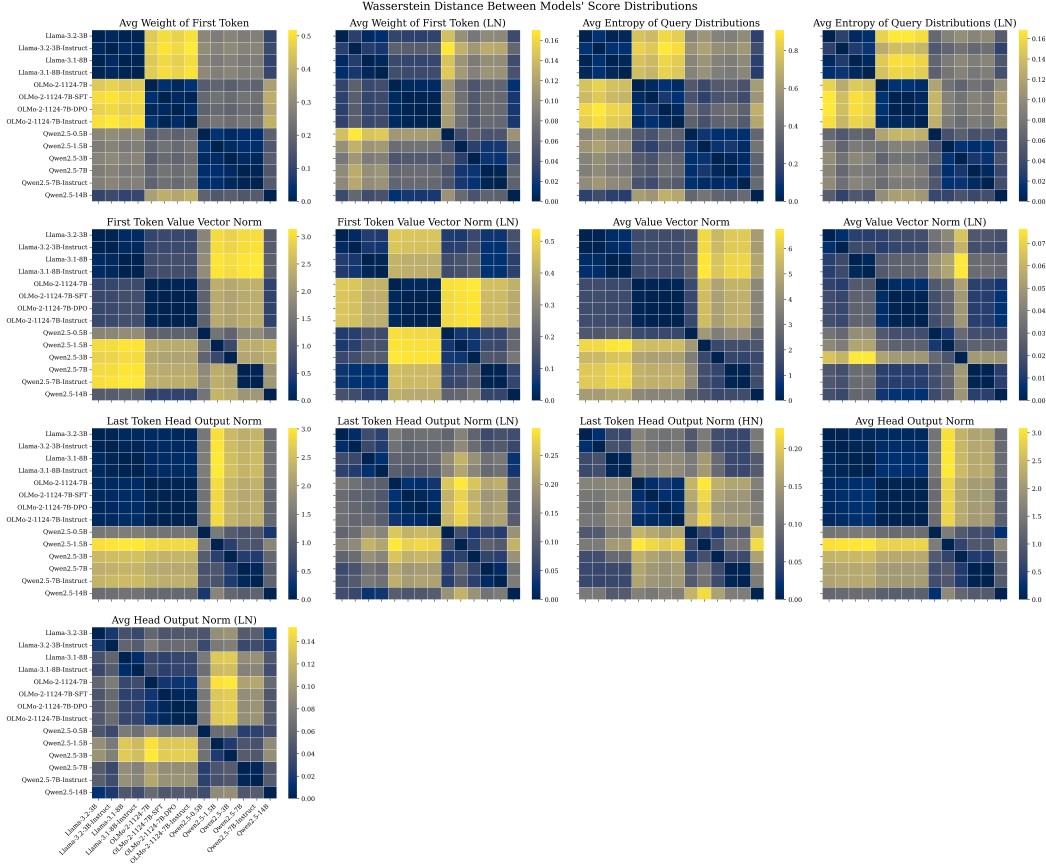

Figure 17: **Wasserstein Distance of Head Score Distributions between Models.** For each of 13 score functions, we plot a matrix of Wasserstein distances between score distributions of different models.

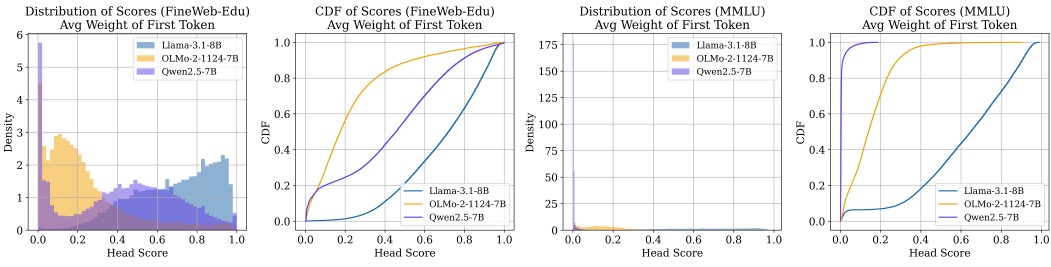

Figure 18: **Distribution of Avg Weight of First Token Scores changes when the input distribution changes.** Using the same 3 models, we find that the distribution of Avg Weight of First Token scores varies significantly when using input sequences from MMLU (right) instead of FineWeb-Edu (left).

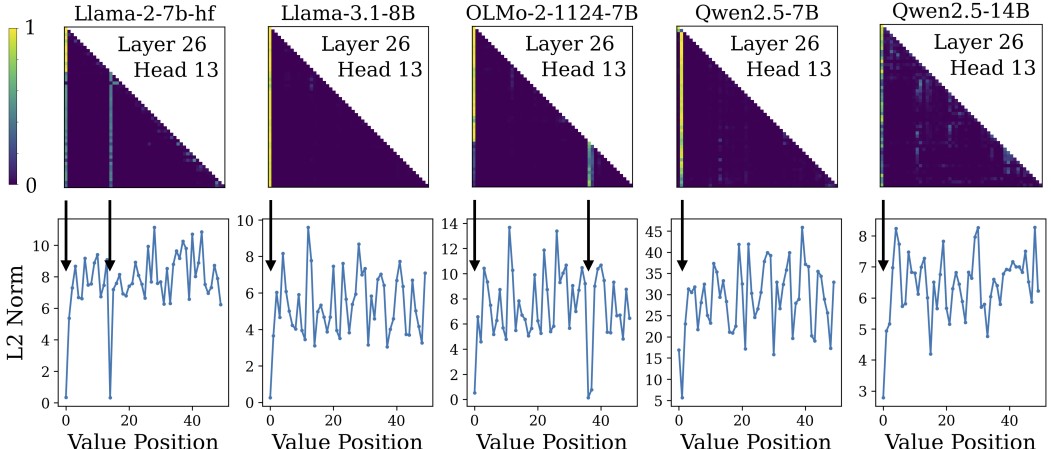

Figure 19: **Attention sink tokens have the smallest value vector norms**. For different models, we input the same question from MMLU. In the first row, we plot the attention weights for an arbitrary head (Layer 26, Head 13) across all models. In the second row, we plot the $\ell_2$-norm of the value vector at every position.

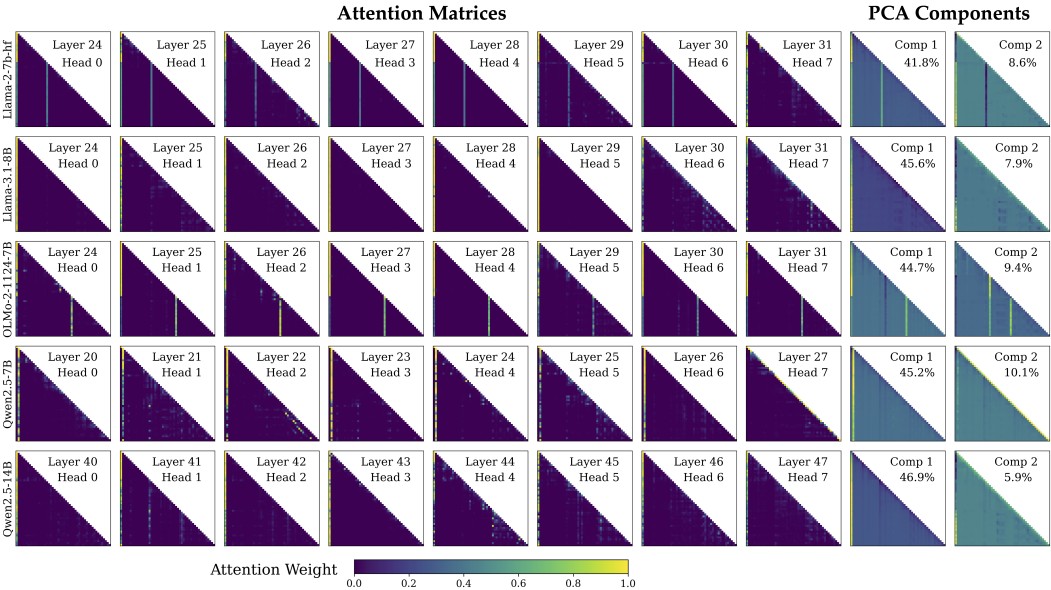

Figure 20: **PCA of Attention Matrices**. Attention patterns from different heads appear homogeneous and dominated by attention sinks. We input the same multiple-choice question from MMLU into Llama-2-7b-hf, Llama-3.1-8B, OLMo-2-1124-7B, Qwen2.5-7B, and Qwen2.5-14B then visualize the attention matrices. Each model row displays a sample of eight attention matrices from the last eight layers, followed by the top-2 principal components of all attention matrices, with the explained variance of each component displayed. The top-2 principal components display bright columns of attention sinks while capturing $\geq 50\%$ of the variance. "L26 H2" denotes the attention head at index 2 of layer 26.

