# OpenReview forum: "Identifying and Evaluating Inactive Heads in Pretrained LLMs"
_ICLR.cc/2026/Conference — ICLR 2026 Poster_

### Official Review · Reviewer_DGdn · 2025-10-25

**Soundness:** 2
**Presentation:** 3
**Contribution:** 2
**Rating:** 4
**Confidence:** 3

**Summary:**

This paper proposes a comprehensive perspective for detecting inactive heads in Transformers and identifies Average Head Output Norm (AHON) as the most effective indicator, thereby enhancing our understanding of inactive attention heads.

**Strengths:**

1.	The paper is well organized and easy to follow.
2.	The conclusion is clear and direct, effectively demonstrating that the Average Head Output Norm (AHON) serves as a reliable indicator for identifying inactive heads.

**Weaknesses:**

1.	While the paper arrives at a clear and well-supported conclusion, it lacks further discussion on how the proposed detection method could be effectively utilized in practice. This limitation raises questions about the extent of the paper’s contribution beyond.
2.	The experiments are conducted on a limited set of datasets, which may restrict the generalizability of the findings to broader domains or tasks.
3.	Although the proposed set of metrics is comprehensive, the paper lacks deeper analysis and interpretation of the observed results. The integration of these indicators might not yet be sufficiently thorough, and the underlying causal explanations for the findings remain somewhat unclear.

**Questions:**

1.	Could the authors further clarify the significance of dynamically detecting inactive heads?
Based on my understanding, this dynamic pruning is not directly applicable for inference efficiency, since the pruning is not static.
If the main goal of the paper is to enhance our understanding of inactive attention heads in Transformers, what is the broader significance of detecting them dynamically? What insights or implications can we gain from this beyond efficiency?
2. I am also curious whether the set of dynamically inactive heads changes significantly with different levels of context or question difficulty？
3. Could the authors comment on whether the presence of “inactive” heads should be viewed as a normal characteristic of the model rather than a deficiency to be mitigated? For instance, in simpler contexts, certain heads might naturally remain unused, whereas in more complex inputs, more heads could become activated. Do the authors expect such adaptive activation behavior, and would they consider it a potentially desirable property?

---

> ### Author Response · Authors · 2025-11-20
> **Addressing feedback**
>
> Thank you for your thorough review. We address your feedback below:
>
> > lacks further discussion on how the proposed detection method could be effectively utilized in practice
>
> We discuss this a bit at the end of Sec. 2, but we agree we can do more. We’ve added App. A.15 “​​Practical Implications and Limitations”. It should be possible to dynamically prune different heads on the fly whenever we identify a head as inactive. But the question is whether the overhead in identifying these heads (by computing scores) will make inference so slow that the compute savings are not worth it. Another potential direction is KV cache compression: specifically, not storing K/V for heads that are inactive. Our work also opens up new directions for investigation like whether one can finetune inactive heads.
>
> > experiments are conducted on a limited set of datasets
>
> We added additional experiments on PIQA (0-shot) and Winogrande (5-shot), following the same experiments as Sec. 4.3. On PIQA, our score functions show that more than 21% of attention heads can be zeroed while keeping average accuracy within 1% of the original model (Tab. 4). On WinoGrande, more than 14% of attention heads can be zeroed while keeping average accuracy within 1% of the original model (Tab. 5). Thus, our claim in the main body of the paper is a conservative estimate of inactive heads. In all cases, our score functions can identify more heads than the prior work of Avg Weight of First Token (AWFT).
>
> > paper lacks deeper analysis and interpretation of the observed results…I am also curious whether the set of dynamically inactive heads changes significantly with different levels of context
>
> We added additional analysis in a new App. A. 5 and Fig. 12 where we plot % of inactive heads per layer, for different kinds of inputs (MMLU vs PIQA vs WinoGrande). We find there is less variance among datasets when using Average Head Output Norm (LN) as the score function, compared to prior work (AWFT), demonstrating the consistency of our score function across data distributions. In OLMo-2, middle layers tend to have fewer inactive heads.
>
> We also added analysis of how inactive heads change as we add tokens from a FineWeb-Edu sequence in App. A.6 and Fig. 13. In measuring both AWFT and AHON (LN) inactive heads, it is clear that inactivity is context-dependent. This property can also be observed in App. A.5 where inactive heads by layer vary with the input distribution. In summary, inactivity appears to be task-specific (A.5, Fig. 12) and context-dependent (A.6, Fig. 13), which lines up with prior work [1].
>
> With this in mind, our main finding is that our AHON (LN) score function can identify a larger fraction of inactive heads than prior work, which only focused on attention weights, and does so in a way that is less model/dataset-dependent. Our work shows that relying on only attention weights for determining inactivity can be misleading because there is a larger set of inactive heads we can find.
>
> [1] Active-Dormant Attention Heads: Mechanistically Demystifying Extreme-Token Phenomena in LLMs, Guo et al., 2024
>
> > dynamic pruning is not directly applicable for inference efficiency, since the pruning is not static…what is the broader significance of detecting them dynamically?
>
> Dynamic pruning has been used for inference efficiency in the past, for example in the case of image classifiers [2,3]. Future work might be able to train policies for selecting heads to execute.
>
> The broader significance of our work is demonstrating that inactive heads are a widespread model behavior, present across model families. This would not have been evident if one relied on a first token attention sink definition like AWFT because not all models use a first token sink (See Qwen2.5-7B, Fig. 19). For insights beyond efficiency: the fact that AWFT performs reasonably well as a score function (Fig. 3) suggests that attention sinks can be indicative of inactive heads. But importantly, there are more inactive heads than heads with attention sinks.
>
> [2] Runtime Neural Pruning. Lin et al., 2017 (NeurIPS)
>
> [3] BlockDrop: Dynamic Inference Paths in Residual Networks. Wu et al., 2018 (CVPR 2018)

---

> > ### Author Response · Authors · 2025-11-20
> > **Responding to questions**
> >
> > > Could the authors comment on whether the presence of “inactive” heads should be viewed as a normal characteristic of the model rather than a deficiency to be mitigated? For instance, in simpler contexts, certain heads might naturally remain unused, whereas in more complex inputs, more heads could become activated.
> >
> > We do believe that some contexts require more “processing” than others. But what we consider to be “hard” for us, may be simple to an LLM. Even if we had a spectrum of easy-to-hard problems, we likely may not see a correlation between hardness and the proportion of active heads (assuming more heads are needed for harder problems). As an example, we think WinoGrande is just as hard as PIQA, but we can identify more inactive heads in PIQA (Tab. 4) than in WinoGrande (Tab. 5). Still, we think there is evidence [2,3,4] that different inputs require different fractions of model parameters and that this is to be expected. Our experiments provide concrete evidence that this occurs in self-attention, and we do believe the sparsity can be beneficial for designing architectures like MoEs but for attention modules.
> >
> > [4] Mixture-of-Depths: Dynamically allocating compute in transformer-based language models. Raposo et al., 2024.
> >
> > We found your suggestions helpful and made a significant effort to address your feedback including multiple paper edits and appendix sections. We would appreciate it if you would consider raising your score based on our response. Let us know if there is more we can do.

---

### Official Review · Reviewer_WQ2Q · 2025-10-26

**Soundness:** 2
**Presentation:** 2
**Contribution:** 2
**Rating:** 2
**Confidence:** 5

**Summary:**

The paper aims to reform the existing attention-sink-based characterization of inactive heads by focusing on the distribution of attention weights in LLMs. It introduces a taxonomy of 13 score functions to identify inactive heads, based on either the attention weight distribution or the norm of an attention head’s value vector or output at different positions, under various normalization strategies. Through ablation-based studies across 14 models, it shows that up to 12% of heads can be ablated without accuracy loss when the proper score metric is chosen, and that output-norm-based metrics outperform traditional attention-weight heuristics in many cases. The study also finds that fine-tuning strategies and moderate scaling barely affect attention behavior, highlighting strong structural invariance across models within the same family.

**Strengths:**

1. This paper represents a meaningful step toward moving beyond the current landscape of inactive head identification and attention pruning, which is predominantly attention-sink-based and focuses heavily on surface-level attention weight characterization. The score metrics proposed in the paper—based on the norms of different aspects of attention head outputs across token positions—are indeed novel. They open the possibility for a new paradigm in this subfield that diverges from the traditional attention-sink-centered narrative and instead evaluates inactivity through more direct and mechanistically grounded measures of an attention head’s functional output.

2. The discussion of intercorrelations among inactive heads defined under different criteria across model sizes, families, and training stages (SFT/RLHF/DPO) is both novel and compelling. This comparative perspective is the paper’s key advantage over prior studies, which typically characterize head inactivity within each model in isolation. By analyzing how these relationships generalize across architectures and alignment methods, the paper contributes a broader, more systematic understanding of attention head behavior.

**Weaknesses:**

1. Despite its novelty, the proposed taxonomoy of inactivity criteria seem highly ad-hoc and disorganized, due to following reasons:

   **Problematic definition**: the author uses a rather unorthodox definition of the attention head output, i.e. as the convex combination of value vectors weighted by attention coefficients prior to multiplication with the output projection matrix. This would result in a $d_v$ (dimension of value embedding) dimensional vector, which diverges from the more widely-accepted and -used circuit-based definition in [1] which multiplies the $d_v$ dimensional vector futher with the corresponding $R^{d_v \times d}$ dimensional slice of the projection matrix.

   **Significant conceptual overlap between criteria**: As a direct consequence of the problematic definition, there is significant conceptual overlap and redundancy between the value-vector-based criteria and head-output-based criteria, since the norm head output at a token position $i$ could be trivially bounded by the maximum (and minimum) norm of value vectors from position 1 to $i$. Thus it is very likely that a head with low average value vector norm will have low average head output norm/last token head output norm and vice versa. Indeed the paper's observation of a 0.55 IOU value between average head output norm and average value vector norm demonstrates this redundancy.

   **No justification for the inclusion and exclusion standards**: The authors made no justifications for why they include/exlucde certain criteria in their taxonomy. For instance, since they included the last token head output norm, they could have easily included the last token value vector norm as well. But instead they only included the first token value vector norm, without explaining the rationale.

   **Disorder related to normalization strategies**: The authors included layer-normalized version of their proposed 6 criteria into the taxonomy, without explaining the reason for this. They further included a head-normalized version specifically for the last token head output norm, but not for the other metrics.

2. The paper spent little efforts in analyzing the properties of the identified inactive heads, which could include the heads distribution across layers, their activation/attention weights patterns for different kinds of inputs, etc. Moreover, the paper only discussed the accuracy consequences of the ablation of the identified heads, without any qualitative analysis of how the ablation affects model outputs, or any fine-grained quantitative analysis of how the ablation affects the model's full forward computation process (hidden states etc). These shortcomings limit the mechanistic depth of the paper's analysis.

3. The writing of certain parts of the paper appears perplexing. For instance, lines 400-403 appear completely irrelevant to the context. There are also grammatical issues here and there. For instance line 420: "Every model and score function combination produce (should be produces)..."

[1] Elhage, Nelson, et al. "A mathematical framework for transformer circuits."

**Questions:**

1, In the first row of Table 2, why is there a difference percentage between AWFT and your method (which is also AWFT)? The same applies to the case of Qwen2.5-1.5B and Qwen 2.5-14B as well.

2, Line 407-408: "Each of 13 score functions use a different set of 7 thresholds, which may not identify 30% of model heads (the maximum we consider)". I am unsure of what this means. Don't you ablate the same percentage of heads (whether it is 30% or not) identified using different criteria?

---

> ### Author Response · Authors · 2025-11-20
> **Addressing feedback**
>
> Thank you for your thorough review. We address your feedback below:
>
>
> > Problematic definition: the author uses a rather unorthodox definition of the attention head output, i.e. as the convex combination of value vectors weighted by attention coefficients
>
> Thanks for bringing up [1]. It’s important we clarify this, so we wrote additional sentences in Sec. 2 + a new App. A.1 to explain why we chose the original definition of a head’s output as the pre-output-projection vector ($d_v$-dimensional) (Vaswani et al., 2017). Importantly, the two definitions we are discussing only affect *identification* not *evaluation* of inactive heads. More specifically, let $Z_i = A_i V_i \in \mathbb{R}^{N \times d_v}$ be the head output for the $i$ head. The circuit definition of [1] defines $Z_i W_O^{(i)}$ as the head output, where $W_O^{(i)} \in \mathbb{R}^{d_v \times d_{\text{model}}}$ partitions the rows of $W_O$ for each head $i$. This doesn’t affect our existing intervention results because whether we choose to define $Z_i$ or $Z_i W_O^{(i)}$ as the $i^{\rm th}$ head's output, when we “zero out a head” $Z_i$, both will be zero.
>
> The question is: does measuring $\||Z_i W_O^{(i)}\||$ help? **We implemented Avg Head Output Norm (AHON) (LN) using this circuit-based definition for 3 models on MMLU and found that it doesn’t make much difference.** Like Tab. 2, below we show the % of heads that can be zeroed while keeping avg acc to within 1% of the original model (higher is better).
>
> | Model                 | Avg Weight of First Token | AHON (LN) | Circuit-based AHON (LN) |
> |-----------------------|---------------------------|---------------------------|-----------------------------------|
> | Llama-3.1-8B-Instruct | 1.01                      | **10.97**                     | 9.88                              |
> | Qwen2.5-3B            | 5.67                      | **7.29**                      | 5.95                              |
> | Qwen2.5-7B            | 1.25                      | 7.54                      | **9.04**                              |
> | Avg of Above Models   | 2.64                      | **8.60**                      | 8.29                              |
>
> > Significant conceptual overlap between criteria…a 0.55 IOU value between average head output norm and average value vector norm demonstrates this redundancy
>
>
> We agree 0.55 IoU is significant, but this is far from a complete overlap, indicating we are still learning something new from evaluating each score function. Despite being similar, in Fig. 3, Average Value Vector Norm is consistently the *worst* score function out of all methods while Average Head Output Norm performs *best* in some cases.
>
> The reason we have so many score functions is to cover a wide-range of simple/obvious features (attn weights, value vectors, head outputs). This gives us a comprehensive picture of what features lead to inactivity.
>
> > No justification for the inclusion and exclusion standards
>
> We designed Sec. 3 score functions based on patterns we observed within a number of attention heads across different model families. We created new A.3 Figs. 7-8 to illustrate specific heads identified by each score function. Let us know if there is more we can do here. There are so many potential simple features and we concede that we cannot be exhaustive.
>
> > Disorder related to normalization strategies
>
> Understood. We included “head normalization (HN)” for completeness (because it was part of our exploration experiments), but we recognize the confusion it may add. We moved it to a new App. A. 2 and removed it from the main text and all figures. Looking at our rank of score functions (Fig. 3), (HN) was always worse than (LN). There was no need to have (HN) in the main body.
>
> > Little efforts in analyzing the properties of the identified inactive heads…the heads distribution across layers, their activation/attention weights patterns for different kinds of inputs
>
> We added additional analysis in a new App. A. 5 and Fig. 12 where we plot % of inactive heads per layer, for different kinds of inputs (MMLU vs PIQA vs WinoGrande). We find there is less variance among datasets when using Average Head Output Norm (LN) as the score function, compared to prior work (AWFT), demonstrating the consistency of our score function across data distributions. In OLMo-2, middle layers tend to have fewer inactive heads.

---

> > ### Author Response · Authors · 2025-11-20
> > **Responding to questions**
> >
> > > Q1: why is there a difference percentage between AWFT and your method (which is also AWFT)?
> >
> > We exclude AWFT (unnormalized) from what we define as “Ours” since it was defined in prior work. We clarify this in the table caption.
> >
> > > Q2. "Each of 13 score functions use a different set of 7 thresholds, which may not identify 30% of model heads (the maximum we consider)". I am unsure of what this means.
> >
> > We revised this sentence to be clearer. In Fig. 4, the thresholds are chosen such that between 0-30% of heads are identified as inactive. But as we describe in Sec. 4.1 “Metrics and Thresholds” we can only *estimate* the proportion to be identified, and because it is not exact, our accuracy curves in Fig. 4 do not always reach 30% model heads zeroed along x-axis (the max x-axis in our plots). The sentence you point to describes why we use a normalized AUC to rank score functions.
> >
> > We found your suggestions helpful and made a significant effort to address your feedback including new experiments, new appendix sections, and other edits. We would appreciate it if you would consider raising your score based on our response. Let us know if there are additional questions we can address.

---

### Official Review · Reviewer_92AY · 2025-10-26

**Soundness:** 4
**Presentation:** 4
**Contribution:** 3
**Rating:** 6
**Confidence:** 3

**Summary:**

This paper investigates inactive attention heads in pre-trained LLMs by proposing a taxonomy of 13 score functions that measure attention weights, value vectors, and head outputs. The authors verify head inactivity through model interventions, zeroing out identified heads and measuring MMLU accuracy across 14 models from 3 families. Their main finding is that more than 12% of attention heads can be zeroed while maintaining accuracy. These results are significantly better than prior methods. The paper also analyzes score distributions to show that finetuning causes minimal changes to attention behavior and that model scaling has little effect until very large scales.

**Strengths:**

1. Clarity and readability
The writing style of this paper significantly enhances the quality of the work.

2. Depth
The paper discusses a wide-range of scores, to identify and evaluate inactive attention heads in a robust manner.

3. Novelty
The paper shares a new surprising finding about LLMs that challanges prior work.

3. Rigorous Experimental Design
The paper tests across 14 models from 3 families (Llama, OLMo, Qwen), it uses model interventions to verify inactive heads rather than just classification.

**Weaknesses:**

1. Single Benchmark Evaluation
The current evaluation is limited to MMLU, but inactive heads might become active in very specific context that are not triggered in MMLU.

2. Lack of Practical Implications
The fact that 12% of heads are inactive is astounding, however, it is unclear what kind of actionable steps can be taken to improve modern LLMs. Does this insight imply a different architecture? Could we remove heads in deployment and achieve better performance?

**Questions:**

1. Do you have a speculation on Why the model does not leverage 12% of its heads?
2. Is it clear how to design better architecture to take advantage of that?
3. Can we prune these heads in production system to improve latency and cost?
4. Do you have any hypothesis on why these head emerge?
5. Can we change the training data or algorithm to incentivize the model to use all attention heads?

---

> ### Author Response · Authors · 2025-11-20
> **Addressing feedback**
>
> Thank you for your thorough review. We address your feedback below:
>
> > Single Benchmark Evaluation The current evaluation is limited to MMLU…inactive heads might become active in very specific context
>
> We added additional experiments on PIQA (0-shot) and Winogrande (5-shot), following the same experiments as Sec. 4.3. On PIQA, our score functions show that more than 21% of attention heads can be zeroed while keeping average accuracy within 1% of the original model (Tab. 4). On WinoGrande, more than 14% of attention heads can be zeroed while keeping average accuracy within 1% of the original model (Tab. 5). Thus, our claim in the main body of the paper is a conservative estimate of inactive heads. In all cases, our score functions can identify more heads than the prior work of Avg Weight of First Token (AWFT).
>
> Inactivity does appear to be task-specific. We added additional analysis in a new App. A. 5 and Fig. 12 where we plot % of inactive heads per layer, and find that inactivity varies for different kinds of inputs (MMLU vs PIQA vs WinoGrande).
>
> > Lack of Practical Implications…insight imply a different architecture?...Q2. Is it clear how to design better architecture to take advantage of that?
>
> We think there is evidence [2,3,4] that different problems require different fractions of model parameters to solve. Our experiments provide concrete evidence that this occurs in self-attention, and we think sparsity can be beneficial for designing architectures like MoEs but for attention modules. We may not need to have all heads in memory if only a subset will be used. A router could potentially be developed to select the appropriate heads. Our work provides necessary background on inactive attention heads so that future work can develop new architectures for this problem.
>
> [2] Runtime Neural Pruning. Lin et al., 2017 (NeurIPS)
>
> [3] BlockDrop: Dynamic Inference Paths in Residual Networks. Wu et al., 2018 (CVPR 2018)
>
> [4] Mixture-of-Depths: Dynamically allocating compute in transformer-based language models. Raposo et al., 2024.
>
> > Q1. a speculation on Why the model does not leverage 12% of its heads?...Q4. Do you have any hypothesis on why these head emerge?
>
> We think some contexts require more “processing” than others, as in [4]. But what we consider to be “hard” for us, may be simple to an LLM, so it may not be the case that seemingly easy problems correlate with more inactive heads.
>
> > Q3. Can we prune these heads in production system to improve latency and cost?
>
> It should be possible to dynamically prune different heads on the fly whenever we identify a head as inactive. But the question is whether the overhead in identifying these heads (by computing scores) will make inference so slow that the compute savings are not worth it. Another potential direction is KV cache compression: specifically, not storing K/V for heads that are inactive. We’ve added App. A.15 “​​Practical Implications and Limitations”.
>
> > Can we change the training data or algorithm to incentivize the model to use all attention heads?
>
> This is an open question we are curious about. As we described in our answer to Q4, inactive parameters may be an inherent consequence of having a model with a fixed compute budget forced to handle inputs of varying complexity (from extremely simple to extremely complex). In its current form, we don’t think data can rid the Transformer of inactive heads.
>
> We found your suggestions helpful and made a significant effort to address your feedback including multiple paper edits and appendix sections. We would appreciate it if you would consider raising your score based on our response. Let us know if there is more we can do.

---

> > ### Comment · Reviewer_92AY · 2025-11-24
> >
> > I thank the authors for their detailed and comprehensive response to the issues raised in my review. I have carefully considered the rebuttal and the revisions made to the manuscript.
> >
> > I am satisfied that the authors have successfully addressed my primary concerns. Consequently, I am happy to raise my score to reflect the improved quality and clarity of the work.

---

### Official Review · Reviewer_8PXa · 2025-10-29

**Soundness:** 3
**Presentation:** 2
**Contribution:** 3
**Rating:** 4
**Confidence:** 3

**Summary:**

This paper investigates the phenomenon of "inactive" attention heads in large language models (LLMs). The authors argue that prior work, which defined inactivity solely through attention patterns (like "attention sinks" on the first token), provides an incomplete picture. They propose that a head can be inactive in multiple ways, not just through its attention weights, but also through its value vectors or its final output vectors.

**Strengths:**

The paper moves beyond a single definition of inactivity to propose a multi-faceted taxonomy. The 13 score functions provide a much richer analysis.

Testing across 14 models from 3 different families provides robust, generalizable results that are not specific to a single architecture or training corpus.

The analysis of score distributions to study the effects of fine-tuning and model scaling provides valuable insights into model behavior.

**Weaknesses:**

**Limited Task Evaluation**: The primary evaluation is on MMLU, a multiple-choice knowledge-based benchmark. The generality of the findings to other tasks (e.g., code generation, long-context reasoning, creative writing) is not verified. A head inactive on MMLU might be active on a different task.

**Static, Single-Pass Analysis**: The analysis is performed on a single forward pass. The role of a head might be context-dependent or dynamically change across layers or tokens in a way that this static analysis misses. The paper acknowledges this by noting that inactivity is verified "in specific contexts."

**Threshold Reliance**: The method relies on choosing thresholds, which, while systematically swept, is still a heuristic. The "inactive" label is not absolute but relative to a chosen threshold.

**Questions:**

Did the authors observe any cases where a head identified as inactive on MMLU became critical for performance on a different type of task? How task-specific do you believe the phenomenon of head inactivity is?

The analysis provides a "snapshot" of head activity. Did the authors investigate if the inactivity of a head is static, or can a head dynamically switch between active and inactive states depending on the specific input token or the evolving context within a sequence?

---

> ### Author Response · Authors · 2025-11-20
> **Addressing feedback**
>
> Thank you for your review. We address your feedback below:
>
> > Limited Task Evaluation: A head inactive on MMLU might be active on a different task
>
> Yes, in A.9 Fig. 18, we compare MMLU inactive heads to FineWeb-Edu inactive heads and find that Avg Weight of First Token scores are data dependent. We also added additional experiments on PIQA (0-shot) and Winogrande (5-shot) in App. A.5 and Fig. 12 where we plot % of inactive heads per layer, for different kinds of inputs (MMLU vs PIQA vs WinoGrande). Again, we see that the dataset affects inactive head proportions in some layers more than others. Notably, our Avg Head Output Norm (LN) score fn is less data dependent (as shown by closer overlapping lines in Fig. 12).
>
> > Static, Single-Pass Analysis: analysis is performed on a single forward pass. The role of a head might be context-dependent or dynamically change
>
> We focus on analyzing a single forward pass because that is how log-likelihood evaluations of MMLU, PIQA, WinoGrande and other MC benchmarks are done. Analyzing generation is complex because zeroing out a single incorrectly identified head can ruin the rest of the generated sequence (i.e. more opportunities for failure): for example, if we zero out one too many heads, we may sample a token representing a different language. We wanted to first understand the simplest case: one forward pass.
>
> Still, we added analysis of how inactive heads change as we add tokens/context from a FineWeb-Edu sequence in App. A.6 and Fig. 13. In measuring both AWFT and AHON (LN) inactive heads, it is clear that inactivity is context-dependent. This property can also be observed in App. A.5 where we analyze inactive heads by layer when varying the input distribution (MMLU vs. PIQA vs. WinoGrande).
>
> > Threshold Reliance: relies on choosing thresholds
>
> When we started analyzing the prior work of Average Weight to First Token (AWFT), this reliance on thresholds bothered us too. In App. Fig. 16, in the CDF for AWFT, we see that for *any* AWFT threshold, different fractions of heads are classified as inactive. In other words, there are wildly different model proportions depending on the threshold. Our normalized score functions fix this (as represented in the overlapping lines of CDFs in App. Fig. 16). For example, in the score distributions for AHON (LN), we see that across model families, **we can use the same threshold** to identify proportions of inactive heads. More specifically, we find in A. 5 that a threshold of 0.455 works well across the board for our AHON (LN) score function.
>
> > Q1 and Q2: How task-specific do you believe the phenomenon of head inactivity is? Can a head dynamically switch between active and inactive states depending on the specific input token or the evolving context within a sequence?
>
> Inactivity appears to be task-specific (A.5, Fig. 12) and context-dependent (A.6, Fig. 13). This lines up with prior work [1] where a head is shown to be active or inactive depending on whether the input is Wikipedia or GitHub. This would also be an example of a head that would be considered critical for one task but not another.
>
> In this context, our main finding is that our AHON (LN) score function can identify a larger fraction of inactive heads, and does so in a way that is less model/dataset-dependent. Our work shows that relying on only attention weights for determining inactivity can be misleading because there is a larger set of inactive heads we can find.
>
> [1] Active-Dormant Attention Heads: Mechanistically Demystifying Extreme-Token Phenomena in LLMs, Guo et al., 2024
>
> We found your suggestions helpful and made a significant effort to address your feedback including multiple paper edits and appendix sections. We would appreciate it if you would consider raising your score based on our response. Let us know if there is more we can do.

---

> > ### Comment · Reviewer_8PXa · 2025-11-26
> >
> > I thank the authors for their response. My concerns have been largely addressed, and I have therefore increased my score.

---

### Official Review · Reviewer_j42Y · 2025-10-31

**Soundness:** 3
**Presentation:** 4
**Contribution:** 3
**Rating:** 8
**Confidence:** 4

**Summary:**

This paper studies the phenomenon of "inactive heads" in pretrained LLMs. To comprehensively identify inactive heads, the authors propose 13 score functions related to attention heads for evaluating the importance of heads and assess their effectiveness in identifying inactive ones. Extensive experiments on MMLU and pretraining corpora show that the proposed "Average Head Output Norm" metric performs best in identifying inactive heads. It can remove as many attention heads as possible while maintaining the original performance, and it detects more inactive heads than the traditional method based on the first-token attention sink. The paper also uses these metrics to briefly analyze attention behaviors across models of different sizes and training settings, showing that finetuning hardly changes attention behavior, while larger models exhibit distinct attention patterns.

**Strengths:**

1. The proposed scoring functions comprehensively covers various aspects of the attention layers, and are extensively evaluated on various LLMs across different model families, showing strong generality.

2. The proposed AHON metric effectively identifies a more complete range of inactive heads and provides useful guidance for studying inactive heads.

3. The paper is clearly written, includes complete experimental results, and is easy to follow.

**Weaknesses:**

1. The main contribution of the paper is limited to identifying inactive heads in LLMs. Beyond that, the discussion on the mechanisms and behaviors of inactive heads is quite restricted. I would like to see why AHON can better identify inactive heads from an internal, model-based behavior perspective.

2. In the experiment section, the paper only uses the pretraining dataset and the 5-shot MMLU dataset. As MMLU is indeed commonly used to evaluate general model capability, using only one dataset is rather limited. Moreover, the 5-shot setting may interfere with the evaluation of inactive heads related to in-context learning.

**Questions:**

1. In Section 4.2, the paper states that different score functions identify different groups of inactive attention heads, with low overlap among them. Could the authors explain what kinds of attention heads are identified by each metric? Do these metrics capture different aspects of the model's behavior?

2. Head scores computed by functions in Table 1 require computation to obtain during the actual inference. I am curious whether the model's static parameters might already contain some clues about inactive heads (e.g., the value projection matrix of each head, or the corresponding output projection matrix).

3. In Section 4.4, Line 453, the authors suggest that larger models learn to specialize their heads in different ways. However, the score distribution change only appears from 7B to 14B, and there is no larger model to support this claim. I hope the authors can validate this hypothesis on larger models (e.g., 32B).

4. Do the findings in this paper have some pratical contributions, such as guiding model performance improvement? I know that this work focuses on interpretability, but it would be better if it also had potential for practical guidance.

5. Minor formatting and typo issues: several elements exceed the page boundary (e.g., Table 1, Figures 6–7, and the example text in Section A.6); the opening quotation mark in Line 148 is incorrect.

---

> ### Author Response · Authors · 2025-11-20
> **Addressing feedback**
>
> Thank you for your thorough review. We address your feedback below:
>
> > discussion on the mechanisms and behaviors of inactive heads is quite restricted. I would like to see why AHON can better identify inactive heads from an internal, model-based behavior perspective.
>
> Regarding inactive head behaviors: we added additional analysis in a new App. A. 5 and Fig. 12 where we plot % of inactive heads per layer, for different kinds of inputs (MMLU vs PIQA vs WinoGrande). We find there is less variance among datasets when using Average Head Output Norm (LN) as the score function, compared to prior work (AWFT), demonstrating the consistency of our score function across data distributions. In OLMo-2, middle layers tend to have fewer inactive heads. We also added analysis of how inactive heads change as we add tokens from a FineWeb-Edu sequence in App. A.6 and Fig. 13. In measuring both AWFT and AHON (LN) inactive heads, it is clear that inactivity is context-dependent. This property can also be observed in App. A.5 where inactive heads by layer vary with the input distribution. In summary, inactivity appears to be task-specific (A.5, Fig. 12) and context-dependent (A.6, Fig. 13), which lines up with prior work [1].
>
> Regarding why AHON can better identify inactive heads: As we mention in Sec. 3 “Head Outputs”, the reason has to do with the fact that AHON best aligns with our definition of an inactive head. We define an inactive head as one that we can remove without affecting accuracy. Because the Transformer uses a residual stream, if the self-attention module’s output is zero, we effectively “skip” the module. Thus, it makes sense to look for head outputs that are small. Or, in the case of AHON (LN), smaller than the layer’s average. Note that just because head outputs are small/near-zero doesn’t mean that removing these will not affect model accuracy, so we had to empirically evaluate it.
>
> [1] Active-Dormant Attention Heads: Mechanistically Demystifying Extreme-Token Phenomena in LLMs, Guo et al., 2024
>
> > paper only uses the pretraining dataset and the 5-shot MMLU dataset…the 5-shot setting may interfere with the evaluation of inactive heads
>
> We added additional experiments on PIQA (0-shot) and Winogrande (5-shot), following the same experiments as Sec. 4.3. On PIQA, our score functions show that more than 21% of attention heads can be zeroed while keeping average accuracy within 1% of the original model (Tab. 4). On WinoGrande, more than 14% of attention heads can be zeroed while keeping average accuracy within 1% of the original model (Tab. 5). Thus, our claim in the main body of the paper is a conservative estimate of inactive heads. In all cases, our score functions can identify more heads than the prior work of Avg Weight of First Token (AWFT).
>
> > Q1. Could the authors explain what kinds of attention heads are identified by each metric?
>
> We created new A.3 Figs. 7-8 to illustrate specific heads identified by each score function. We also provide a summary of the kind of head that each score function is meant to capture and provide a short explanation for each. We designed Sec. 3 score functions based on patterns we observed within a number of attention heads across different model families.
>
> > Q2. I am curious whether the model's static parameters might already contain some clues about inactive heads (e.g., the value projection matrix of each head, or the corresponding output projection matrix)
>
> We are actually actively investigating this! Given that the AWFT score function does reasonably well (Fig. 3), it may be possible to analyze Q/K projections and determine whether attention sinks are more likely.
>
> > Q3. Section 4.4, Line 453, the authors suggest that larger models learn to specialize their heads in different ways. However, the score distribution change only appears from 7B to 14B
>
> The specialization can also be seen when measuring AWFT (LN) scores on models going from Qwen2.5-0.5B to 14B (Fig. 5, Right), where score distributions become more and more different (larger Wasserstein distance).
>
> > Q4. Do the findings in this paper have some pratical contributions, such as guiding model performance improvement?
>
> We discuss this a bit at the end of Sec. 2, but we agree we can do more. We’ve added App. A.15 “​​Practical Implications and Limitations”. It should be possible to dynamically prune different heads on the fly whenever we identify a head as inactive. But the question is whether the overhead in identifying these heads (by computing scores) will make inference so slow that the compute savings are not worth it. Another potential direction is KV cache compression: specifically, not storing K/V for heads that are inactive. Our work also opens up new directions for investigation like whether one can finetune inactive heads.
>
> We also fixed formatting/typos you raised. Thank you for helping make our paper better.

---

> > ### Comment · Reviewer_j42Y · 2025-11-27
> >
> > Thank you for the response. I appreciate the effort the authors have put into improving the paper, and my concerns have been largely addressed.
> > Still, regarding the reply to Q3, I still hold that current experiments are not sufficient to support the claim (larger models learn to specialize their heads in different ways): in Fig. 5 Right, the model sizes of 1.5B, 3B, and 7B increase in order, but their score distributions remain quite similar. If computation resources prevent further validation on a 32B model, I suggest the authors at least running the same experiments on Qwen3 models of different sizes to strengthen the claim.
> >
> > Overall, considering that this work indeed proposes a decent solution for detecting inactive heads, I decide to maintain my current positive score.

---

### Author Response · Authors · 2025-11-20
**Summary of Rebuttal Improvements**

We’d like to thank every reviewer again for their time. We read each review carefully and made a number of improvements to the paper, which we summarize below:
1. Added datasets: PIQA (0-shot) and Winogrande (5-shot) in Figs. 10-11 and Tab. 4-5 within A.4
2. Added analysis for multiple-forward-passes in A.6 Fig. 13
3. Added analysis for % of inactive heads per layer in A.5 Fig. 12
4. Added example attention heads identified by each score function in A.3 Figs. 7-8
4. Explanation for standard vs circuit-based head output definition in A.1
5. Explanation of practical implications and limitations in A.15

And other more minor changes. We believe these additions have improved the work, and we invite reviewers to consider the main insight of our paper: We find that heads exhibiting attention sinks (as measured through Avg Weight to First Token (AWFT)) are not necessarily the ones that can be removed, and that score functions measuring head output norms serve as a better metric for identifying inactive heads.

---

### Author Response · Authors · 2025-12-01
**Summary of Rebuttal Discussion**

Dear AC, in response to ICLR’s policy change on 11/28, we want to summarize the discussion period for our paper. Replies from reviewers who responded to our rebuttal were clearly positive:
- Reviewer j42Y maintained **8**
- Reviewer 92AY increased from 6 to **8**
- Reviewer 8PXa increased from 4 to **6**

And while the remaining two reviewers did not respond, we are confident that we adequately addressed their concerns, which we describe below:
- Reviewer WQ2Q is primarily concerned with the definition of an attention head output, overlap between score functions, and disorder in normalization strategies. In response, we clarify why we use the standard attention head output definition front-and-center in a new section A.1 “Defining an Attention Head Output”, we run new experiments on a circuit-based head output definition, and we show that it does not improve results over AHON (LN) on average. We also explain our spectrum of score functions and simplify normalization (by moving additional normalization strategies to Appendix).
- Reviewer DGdn is primarily concerned with practical implications, limited datasets, and analysis of observed results. In response, we wrote a new section A.15 “​​Practical Implications and Limitations”, added results for two new datasets, and added analysis of multiple-forward-passes in A.6 Fig. 13 and of percent of inactive heads per layer in A.5 Fig. 12.

We are thankful for the feedback we received, as it led to multiple paper improvements. We believe the positive reception by reviewers during rebuttal reflects “the improved quality and clarity of the work” (Reviewer 92AY).

---

### Meta-Review · Area_Chair_7cda · 2026-01-06

**Summary:**

## Summary
This paper studies the phenomenon of inactive heads in large language models: the heads that can be zeroed out without hurting models’ performance. Previous results focused on the “attention sinks” phenomenon and proposed to use Avg Weight of First Token (AWFT) to identify inactive heads. The paper proposes a total of 12 scores, each identifying different sets of candidates for inactive heads, and finds that 1) Avg Head Output Norm (AHON) is useful for identifying heads that can be zeroed out, that 2) around 12% of heads can be zeroed out while keeping average accuracy within 1% of the original model. The authors also compare score distributions of different models to analyze which models are closer to each other. The key findings are: 1) finetuning strategies (STF, DPO, RLHF) have almost no effect on score distributions, and 2) score distributions are not very sensitive to model scaling, but the distributions change when models are scaled further.

## Reviewer Concerns
Major concerns raised by reviewers can be summarized as follows.
- **Using only MMLU**. Most reviewers raised concerns that the main experiments were only carried out on a single dataset: MMLU. This raises concerns that the conclusions drawn from this dataset may not apply to other contexts/benchmarks.
- **Lack of deeper analysis**. Reviewers DGdn, WQ2Q, and j42Y pointed out that the paper is limited to just identifying inactive heads and does not go beyond it. The authors do not analyze mechanisms, behaviors, and/or distributions of such inactive heads.
- **Lack of practical applications**. Reviewers j42Y, 92AY, and DGdn pointed out that the paper lacks practical implications, although the fact that 12% of the heads are dormant is interesting.
- **Ad-hoc nature**. Reviewer WQ2Q criticized that 1) the paper uses an unorthodox definition of head output, 2) some criteria overlap significantly, 3) no justifications for including/excluding certain criteria, and 4) LN and HN versions are not justified.
- **Single-pass analysis and threshold reliance**. Reviewer 8PXa criticized that 1) the paper only considers single-pass analysis (i.e., no multi-pass scenarios such as autoregressive generation) and 2) relies on manually choosing thresholds.

**Reviewer Concerns:**

Unfortunately, the discussion period closed before we received feedback from two out of the five reviewers. Based on my reading, the authors’ responses can be summarized as follows.

- **Using only MMLU**. The authors added additional experiments on PIQA (0-shot) and Winogrande (5-shot). The inactive head ratios were even higher for these datasets. AHON and AHON (LN) were superior on these datasets too, although the gaps seem narrower.

- **Lack of deeper analysis**. The authors added Appendices A.5 and A.6 to include deeper analyses, including the distribution of inactive heads for different tasks and behaviors of the scores as context gets longer.

- **Lack of practical applications**. The authors added Appendix A.15 to discuss this matter, where they point to dynamic head pruning (although it is unlikely given the definition of AHON, as authors also mentioned) and KV cache compression as potential applications. In terms of architectural innovation, the authors suggest adding routers to attention modules.

- **Ad-hoc nature**.
  - The authors clarified the definition of head output and added an appendix for clarification.
  - The authors agreed that 0.55 IoU is significant but pointed out that the scores have different performance in identifying heads.
  - The authors added attention pattern examples to visualize different roles of score functions.
  - The authors decided to remove the head-normalized version, which was included as the 13th score function in the original submission, from the main text altogether. The authors did not specifically justify adding LN versions. This point may require further motivation, as Reviewer WQ2Q mentioned.

- **Single-pass analysis and threshold reliance**. The authors clarified the difficulty of considering multi-pass tasks, and emphasized that single-pass analysis is considered as the starting point. As for the threshold, they pointed out that although having to choose thresholds is not ideal, AHON (LN) has quite score distributions consistent over different models and thresholds can be shared across different models.

**Reviewer Scores:**

The initial reviews had scores 8/6/4/4/2. Reviewer j42Y (initial score 8) responded to the rebuttal and maintained their score, while Reviewers 92AY (initial score 6) and 8PXa (initial score 4) raised their scores by 2. Here is what I expect about the reviewers’ final assessment, focusing on the negative reviewers:
- I believe many concerns raised by **Reviewer WQ2Q** (score 2) were handled properly. The one that potentially remains unresolved is Weakness 1: ad-hoc and disorganized nature. I believe that the reviewer’s criticisms based on the high IoU and inclusion/exclusion of certain scores comes from a slight misnomer that the paper uses: taxonomy. To me, taxonomy is more of a “classification into disjoint buckets” where the buckets need to form a full partition of the whole. However, what the paper is actually doing is to try out different score functions and corresponding sets of heads (where overlaps are allowed) to identify a better score function. This mismatch is probably why the high overlap and exclusion of other potentially good candidates may come as disturbing to some readers. I hence recommend deleting any use of “taxonomy” from the paper and reposition the paper as a search for a better score function.
- As for **Reviewer DGdn** (score 4), although I am not sure about the practical applications part (Weakness 1), I believe that the “single-dataset” and the “deeper analysis” parts (Weaknesses 2 & 3) were addressed by the authors. The reviewer may or may not have raised their score.

The paper has some limitations, such as limited practical applications. However, I believe that the paper has meaningful contributions shedding light on the understanding of inactive attention heads. I therefore recommend acceptance of this submission. I strongly encourage that the authors incorporate the discussion into the paper, and also add more “in-depth analysis” into the main text.

---

### Decision · Program_Chairs · 2026-01-26

Accept (Poster)